

# Mountain water cellars: a chemical characterization and quantification of the hydrological processes and contributions from snow, glaciers and groundwater to the Upper Mendoza River basin (~ 32 ° S), Argentina

Sebastián A. Crespo [1,2], Julieta N. Aranibar [2, 3], Francisco Fernandoy [4,5], Leandro Cara [2]

[1]Instituto de Geografía, Facultad de Ciencias del Mar y Geografía, Pontificia Universidad Católica de Valparaíso, Valparaíso, 2362807, Chile
[2]Instituto Argentino de Nivología, Glaciología y Ciencias Ambientales, Conicet, Mendoza, CP5500, Argentina
[3]Facultad de Ciencias Exactas y Naturales, Universidad Nacional de Cuyo, Mendoza, CP5500, Argentina
[4]Laboratorio de Análisis Isotópico LAI, Facultad de Ingeniería, Universidad Andrés Bello, Viña del Mar, 2531015, Chile
[5] Centro de Investigación para la Sustentabilidad CIS, Facultad de Ecología y Recursos Naturales, Universidad Andrés Bello, 8370251, Santiago, Chile

*Correspondence to*: Sebastián A. Crespo (sebacrespo.oliva@gmail.com)

**Abstract.** Between 2010 and 2015 the Central Andes of Chile and Argentina (32–37°S) suffered the effects of a mega drought without precedents in the instrumental period, where 71% of weather stations showed more than 30% of rainfall scarcity in Central Chile. The Cordillera Principal geological province, in the Upper Mendoza River basin, receives almost exclusively winter precipitation originated from the Pacific Ocean moisture. In addition to the snow that precipitates in this area of 3023 km$^2$, there are 951 ice bodies, covering an area of 404 km$^2$. The Mendoza River flow provides fresh water for more than 1.1 million inhabitants in this agriculture based arid region. Given the high inter–annual variability of snowfall, strongly affected by ENSO events, and the aridity of the region, it is crucial to quantify the contribution from different water sources to the Mendoza River flow. Glaciers play an important role regulating water availability, with mass accumulation in wet and cold years, and melting in hot, dry years. Understanding their dynamics as a function of environmental variables will help us predict water availability under a changing climate. Combining the instrumental record of streamflow from glaciers and rivers, meteorological data, remote sensing of snow covered area and chemical analysis of different water sources, this study attempts to understand climatic variables that control thawing, and the hydrological contribution of different glaciers to the streamflow during a dry period. Isotopic composition allowed us to differentiate snowmelt from glacier ice melt. In addition, it was possible to detect contributions of summer rainfall from Atlantic origin, in their unique storms that reach the Cordillera Principal, even when they had not been registered at weather stations. Finally, with end member mixing analysis, the relative contribution from different water sources were quantified over time, showing the temporally increasing contribution of glacial and periglacial environments as the melting season progresses.



# 1 Introduction

The water supply for the oasis irrigated by the Mendoza River, in the northern part of the Mendoza Province, Argentina, has been under extraordinary pressure following the mega–drought that affected the Central Andes during the period 2010–2015 (CR[2], 2015; Cornwell et al., 2016; Gonzalez–Reyes et al., 2017). This basin is supplied mainly from snow contribution in years of normal to abundant loads (Masiokas et al., 2006), while from glacial and periglacial sources during drought events. Because of the limitations given by the climate and topography hold by one of the largest mountain ranges on the planet (Cornwell et al., 2016), these contributions have not been fully determined in the Central Andes. Without this knowledge, any input change from snow to ice calculation with models may be over– or under–estimated (Rodriguez et al., 2014). About 70% of the waters of this basin are generated in the geological province of Cordillera Principal (DGI, 2006a) which in turn presents ideal conditions for the study of water delivered by glaciers, snow and groundwater due to the null forest cover, low cloud cover and infrequent rainfall during the melting period (Cornwell et al., 2016).

Different water sources contribution to the Mendoza River flow in the Cordillera Principal geological province, whether rainfall, groundwater, ice bodies or snow melting, depend on different factors, mainly weather conditions (Paterson, 1969; Corripio et al., 2007; Pellicciotti et al., 2008). In this part of the basin, moisture comes almost exclusively from the Pacific Ocean (Viale and Nuñez, 2011; Hoke et al., 2013; Crespo et al., 2017), so that interferences of Atlantic moisture sources are negligible. Although the source of atmospheric moisture (Pacific), and various weather factors that modulate the melting of snow and ice are known (Lliboutry, 1958; Paterson, 1969; Corripio et al., 2007; Pellicciotti et al., 2008), it is difficult to estimate the relative contribution of snow, ice and groundwater, as they vary temporally, according to the dynamics of winter accumulation and melting in each thawing season. The different responses of ice bodies and snow to weather conditions also depend on the intrinsic properties of each solid water source, such as the albedo, density and thickness of snow or ice cover and the amount and type of sediment contained in glaciers (Lliboutry, 1958; Paterson, 1969; Trombotto and Ahumada, 2005; Corripio et al, 2007; Pellicciotti et al., 2008). Because of the lack of precipitation during the melting period (Cornwell et al., 2016), the albedo of snow and ice remain almost constant, except during the transition from snow to ice (Pellicciotti et al., 2008). Consequently, the river flow is initially derived from snowmelt, followed by the contribution of ice bodies (Crespo et al., 2017). However, this dynamic may vary temporarily and spatially for different basins, making the estimation of the amount and distribution in space and time of the snow pack in each basin more difficult.

The more widely used techniques for ice melt quantification in a glaciated basin (outflow, mass balance or satellite images), are not enough to distinguish the temporal changes of different sources (i.e. from snow to ice, permafrost or groundwater). Differences in water sources origin, linked to sediment and air contact time with water, generate differences in ions and stable isotopes composition (Crespo et al., 2017) in space and time. This behavior provides a natural tracer of flow inputs along the melting season by different water sources and sub–basins to a river. In other similar glaciated basins from different geographical regions like the Bhagirathi River in Indian Himalayas (Lambs, 2000), the use of electrical conductivity and $\delta^{18}O$ composition, served as tracers to identify water from ice, snow and rain water sources. Comparable findings were



published by Lambs et al. (2010) for the Garonne Valley (France), where runoff water from high altitudes was identified using stable oxygen isotopes and conductivity data from river water samples. Similarly, Pu et al. (2013), using only $\delta^{18}O$ composition from different water sources in the Baishui River catchment (China), made an hydrograph temporal separation between rain and melt water contributions to the river, but did not differentiate between snow and ice melting (neither groundwater) contributions separately. Moreover, quantitative estimation of the diverse water sources have been achieved for the Tarim River, Central Asia, by Fan et al. (2016), where a marked seasonal variability was identified by the use of water stable isotopes and electrical conductivity.

The aim of this study was to evaluate the importance of ice bodies on river flow during a dry year, and the environmental factors which control glacial melting. Based on hydrological estimates, glaciers are presumed to maintain basal flow during dry years. Using tracers, we quantified this importance, and understood the major factors that control glacial melting in this region, where glaciers are in risk by global warming and prospective land use. To answer questions like: Which environmental variables control the initial thawing of ice bodies? At which time of the season each of the relative inputs from every water source feeding the main rivers of the Mendoza River basin became relevant? How can these sources be estimated and measured?, we proposed the following related hypotheses: a. The variability of the water flow discharged by different ice body types responds to specific meteorological variables that control ice melting in a significant way. b. Due to differential residence and exchange time with sediments and atmosphere, the different water sources will display different ionic and stable isotope composition. c. The amount of water delivered by each source is reflected in the chemical composition of the river to which they drain.

Summarizing, using meteorological data, streamflow measurements and chemical tracers, the most influential meteorological variables controlling the melting and water delivery by ice bodies and the temporal variation of snow, ice and groundwater contribution to the main rivers of Cordillera Principal mountain range, within a melting season, were studied. A snow contribution at the beginning of the summer, a shift to the ice bodies contribution to the end of summer and a trend to a greater influence of groundwater in autumn to winter, as noted in the period 2011–2012 (Crespo et al., 2017), were expected.

## 2 Methodology

### 2.1 Streamflow and environmental variables analysis

Previous studies (Lliboutry, 1958; Paterson, 1969; Hock, 1998; Brenning, 2003; Francou and Poyaud, 2004; Corripio et al., 2007; Pellicciotti et al., 2008), inferred that the dynamics of water delivered by melting ice bodies respond manly to a few environmental variables, such as precipitation and temperature (Llorens, 2002), controlling hydrological process. Due to both, climate and hydrology data scarcity in glacial basins, flow rates were measured at two representative areas of different glacier types: in Mt. Tolosa rock glaciers conglomerate and at the snout of the Horcones Inferior glacier. Meteorological variables were also measured and analyzed in two stations located in the area of the analyzed cryoforms: one in "Laguna de Horcones" at 3043 m a.s.l. ("1" in Fig. 1) and in "Plaza Francia" ("2" in Fig. 1), at 4016 m a.s.l., on the banks of the



Horcones Inferior Glacier (Fig. 1). Groundwater, snow and mixed contributions from glaciers and snow basins were also analyzed to obtain a chemical characterization of them and contrast with glacier dominated basins.

To infer the melting water flow, ultrasonic limnimeter sensors were installed in the pro–glacial river of the mentioned glacier basins. These sensors measured the height of the streams automatically and continuously. Periodical calibration curves of
height–discharge to infer the flows were developed in field. The installed sensors were Campbell Scientifics, SR50a model, powered by a solar panel with data logger CR200x, also Campbell Scientifics, with 0.25 mm resolution and ± 1 cm sensor accuracy.

### 2.1.1 Height–discharge calibration curves

Two different procedures for each analyzed site were applied for the construction of the calibration curve, which links the
height measured continuously by the sensor and the volumetric flow. One was by constant saline flow (Gordon et al., 2013) and the other by the velocity–area method (Francou and Poyaud, 2004). Selected sites were:

Mt. Tolosa rock glaciers conglomerate: Located between the peak "Leñas del Tolosa" and the "Morro El Paso" peak, a cluster of four rock glaciers is located in the Cordillera Principal geological province, at 32.80°S – 70.01°W. They range from 3509 to 3749 m a.s.l., with an average altitude of 3614 m a.s.l., a mean length of 378 m and a total covered area of 0.16
km$^2$. This site was chosen for the simplicity provided by the cryogenic origin of the rock glaciers covering this sub–basin, accessibility throughout most of the year and because it is near (~ 6 km), and in the same geological province, to the Horcones Inferior Glacier. Furthermore, the similarity in their orientation (south) and average altitude of both glacier basins analyzed reduces the environmental noise when comparing water delivery dynamics of these different kind of ice bodies.

Given its low flow, in the Mt. Tolosa rock glaciers conglomerate, periodic saline tracer constant flow measurements were
performed (Gordon et al., 2013). These dissolution methods involve introducing a tracer substance as a chemical tracer (in this case table salt) in the stream and then monitor concentration changes downstream. This method is especially useful in turbulent mountain streamflow (Moore, 2004; Gordon et al., 2013). The salt solution was injected at a constant flow rate using a Mariotte bottle built for that purpose. Using a digital conductometer, calibration curves were constructed with measurements of salinity downstream, after pouring the saline solution. Eighteen streamflow measurements were made and
adjusted with heights recorded by the sensor according to the limnimetric equation $Qm = (61.27-h)*199.7^{-1}$ ($R^2 = 0.91$); were, $Qm$: measured stream flow and $h$: measured height. The determination coefficient between calculated and observed streamflow was 0.95, for $y = 0.999x - 2E^{-05}$.

Horcones Inferior debris covered glacier: Located in the Aconcagua Mt. Provincial Park (32.73°S and 69.97°W) this basin drains to the Horcones River, a tributary of the Cuevas River, which drains to the Mendoza River (Fig. 1). The debris
covered glacier is distributed from 3472 to 5460 m a.s.l., with an average height of 4151 m a.s.l. It has southeast orientation, a total length of 12.7 km and 6.95 km$^2$ area.

To calculate the Horcones Inferior Glacier streamflow, the gauges were performed by the method of speed–area (Francou and Poyaud, 2004). In this case, weekly gauges were made during the entire summer season. To calculate the area of the





river, profile measurements were made every 30 cm. The flows obtained by continuous measurements with the ultrasonic sensor and those calculated from gauging, adjusted to the equation: $Qm = -0.106*h + 20.44$, for: $Qm$: measured flow and $h$: measured height, with a lineal correlation coefficient $r = -0.95$. The 91% of the variation of the calculated flow could be explained by the model: $Qcalc = 0.902*Qm + 0.362$, where: $Qcalc$: calculated flow.

### 2.1.2 Environmental variables

The Irrigation General Department of Mendoza weather station (labelled as 1 in Fig. 1), located at 3043 m a.s.l. in "Laguna de Horcones" (32.80ºS – 69.95°W), hereinafter "Lagu", measured: air temperature, soil temperature, wind speed and direction, relative humidity, incident radiation and snow water equivalent. Air and soil temperature HOBO sensors were also

installed above the debris covered glacier Horcones Inferior (labelled as 2 in Fig. 1), at 4016 m a.s.l. (32.69°S – 69.97°W), hereinafter as "Francia". Thus, by having two temperature measurements separated by a hypsometric difference of 973 m, it could be established an air and soil temperatures altitudinal gradient for the study area. These meteorological measurements were also used to the study of the Mt. Tolosa rock glaciers conglomerate region.

### 2.1.3 Streamflow and environmental variables statistical analysis

Flow and environmental variables were evaluated in generalized linear models (nlme package, Pinheiro et al., 2013) in the program R (R Core Team, 2013). Flow was considered the response variable, as a function of environmental data (predictor variables).

### 2.2. Natural tracers

Satellite remote sensors indicate snow covered area, but they are insensitive to the thickness of the snow cover.

Meteorological stations can provide precise data, but make extrapolation difficult from their spot measurements. Therefore, to detect changes in contributions from various sources throughout the period of snow and ice melting, the use of chemical tracers (ions and water stable isotopes) was proposed, in addition to weather variables measurements and their links with streamflow dynamics.

Along the melting season 2013–2014, the major rivers of Cordillera Principal were sampled, with a monthly frequency:

Vacas River in Punta de Vacas, Tupungato River in Punta de Vacas, Horcones Superior River and Horcones River (before flowing into the Cuevas River) and Cuevas River in Puente del Inca (Table S1 and Fig. 1). "Confluencia Vieja" and "Confluencia Nueva" springs, located in the Horcones Inferior Glacier region, were also monthly sampled as the waters of the Mendoza River in its initial part, after the confluence of the major tributaries (Cuevas, Vacas and Tupungato rivers).

A weekly sampling was designed for the Cuevas River in Punta de Vacas, the water flows outcoming from the Mt. Tolosa

rock glaciers conglomerate, the waters of the Horcones Inferior Glacier, the snow catchments ("Los Puquios", "Valle Azul"



and "Santa Maria", geothermal waters of "Puente del Inca" (sampled at two sites in the so–called "Champagne Glass" and at the "Old hotel tunnel") and the springs of "Vertiente del Inca" and "La Salada Stream" (Table S1 and Fig. 1).

Totalizers for summer precipitation samples were deployed in the area of "Lagu" and in the Mt. Aconcagua camp "Confluencia", which is located 476 m upper and 6025 m in straight line, at 3433 m a.s.l., from "Lagu" (Fig. 1). The

collectors were PVC pipes (2.5 cm in diameter, 1 m long) with a funnel at the top to improve uptake. They were filled with a light mineral oil inside to prevent evaporation.

The total number of samples taken was 155, in 20 sites, since November 2013 to March 2014. In the case of "Confluencia Vieja" and "Confluencia Nueva" springs samples, Horcones River, Horcones Superior River, and the gauging of the Horcones Inferior Glacier and "Lagu" station measurements, the study period was from December 2013 to March 2014.

Electrical conductivity and water temperature were measured in situ with a digital multiparametric (Thermo Scientific Orion Star A329) and a conductivity electrode DuraProbe 4. Samples for isotopic analyzes were taken in plastic tubes of 15 ml and sealed with a thermoplastic cohesive (PARAFILM®) to prevent evaporation. Samples for ion concentration analysis were collected in 1 l plastic bottles, and cooled to 6°C until analysis.

### 2.3 Lab analysis

The ionic concentration from ice bodies and rainfall samples were analyzed at (IANIGLA–CONICET). These samples were analyzed for: temperature, electrical conductivity, pH and the following major ions (expressed in me l$^{-1}$): bicarbonate ($HCO_3^-$), sulfate ($SO_4^{-2}$), chloride ($Cl^-$), calcium ($Ca^{+2}$), magnesium ($Mg^{+2}$), sodium ($Na^+$) and potassium ($K^+$). Following the methodology of Standard Methods for the Examination of Water and Wastewater (Clesceri et al., 1999), $HCO_3^-$ was analyzed by the volumetric method. $Ca^{+2}$ and $Mg^{+2}$ were determined by volumetric complexometry. Using the spectrometric flame

method, $K^+$ and $Na^+$ were determined, WITH a flame photometer (Metrolab, model 315). For $Cl^-$, electrical conductivity, pH and temperature, a multiparametric (Thermo Scientific Orion Star A329) sensor was used with specific electrodeS for chlorides, electrical conductivity and pH. The $SO_4^{-2}$ ion was analyzed with a Hach spectrophotometer (DR2800) with the 8051 method, SulfaVer4, based on the precipitation of $BaSO_4$ (barium sulfate)

A laser mass spectrometer Picarro L2130–I WS–CRDS (wavelength scanned cavity ring–down spectroscopy) was used to

analyze water stable isotopes at the lab from the Environmental Studies Group (GEA), San Luis Applied Mathematics Institute (IMASL–Conicet). The $\delta^{18}O$ and $\delta^2H$ analytical uncertainty was 0.3 ‰ and 0.5 ‰, respectively. Standardization was based on the Vienna Standard Mean Ocean Water (VSMOW).

### 2.4 Ion concentration and water stable isotopes statistical analysis

In order to identify the main sources of variation in the data, a Principal Component Analysis (PCA, package FactoMineR

1.27, Husson et al., 2014), was applied to ion concentration and stable isotope data. To determine the significance of the variables mostly related to the two main axes of variation identified in the PCA analysis (electrical conductivity and $\delta^{18}O$), generalized linear mixed–effects models (package nlme, Pinheiro et al., 2013) were used. Water source was considered a



fixed factor, with categories corresponding to the hydrological origin: glacier, rock glacier, groundwater, and snow basins. "Debris covered glacier" was considered indistinctly as "glacier" for isotopic composition analysis purpose, because there were no significant differences between uncovered glaciers and debris covered glaciers (Crespo et al., 2017). Linear models were fitted, considering different intercepts (each water source), to compare the different water sources. The sampling sites

were considered as random effects, to avoid pseudo replication due to repeated measurements in the same place over time (Crawley, 2007; Zuur et al., 2009).

## 2.5 Snow covered area analysis

Snow Cover Area (SCA) can be estimated from optical sensors due to the high snow reflectance in the visible Electro Magnetic spectrum and the low reflectance in the infrared zone. The most commonly used index to snow detection for

optical sensors is the Normalized Snow Difference Index (Eq 1) (NDSI) (Riggs et al., 2006). This index is usually calculated at the top of atmosphere (Dietz et al., 2012).

$$\text{NDSI} = \frac{(\text{Visible} - \text{SWIR})}{(\text{Visible} - \text{SWIR})} \quad (\text{Eq. 1})$$

One of the most important optical passive sensors, both for its long time series (data since year 2000 to nowadays) and for its temporal resolution, is MODIS. This offers a daily revisit time for each of its platforms (Aqua & Terra). MODIS sensors deliver daily snow products, with a 500 m spatial resolution. In this way, both satellites allow to obtain complementary information of the terrestrial surface in different hours of the day. The basic algorithm of the NDSI is transformed into a boolean information SCA, using a common threshold for dividing into Snow/No snow areas. In this work a product of level

3 of processing delivered from NSDI has been used (Fig. 10 and 11). Two hundred and seven images were used spanning the period from 9/29/2013 to 5/5/2014 of MOD10A1 product obtained by Terra Sensor, and of MODIS Aqua sensor (MYD10A1 product). Both products were obtained through the Earth Observing System Data and Information System (EOSDIS; http: //reverb.echo.nasa.Gov/). By combining them to get one product through Cara algorithm (Cara et al., 2016), and using a threshold of 30% of cloud coverage, which is the commonly used and agreement threshold for accepting or

discarding information (Roy et al., 2010; Bergeron et al., 2014).

## 2.6 End member mixing analysis (EMMA)

In order to quantify the relative contributions from each water source to the Cuevas River, which is the receptor of the tributaries from the different kinds of water sources analyzed in this work, an end member mixing analysis was carried on (Liu et al., 2004).





Thus, the equation for proportions (f) of contributions to the Cuevas River, which takes into account the contribution of three components (snow or rock glacier, glacier and groundwater) requires the use of two tracers (electrical conductivity and concentration of $\delta^{18}O$ ), with the following system of equations:

$1 = f_1 + f_2 + f_3$   (Eq. 2)

$C_1^1 f_1 + C_2^1 f_2 + C_3^1 f_3 = C_t^1$ (Eq. 3)

$C_1^2 f_1 + C_2^2 f_2 + C_3^2 f_3 = C_t^2$ (Eq. 4)

For:

$f$: discharge fraction

$C$: component concentration

The subscripts refer to the components and superscripts to the tracers

Solutions:

$f_1 = \frac{(c_t^1 - c_3^1)(c_2^2 - c_3^2) - (c_2^1 - c_3^1)(c_t^2 - c_3^2)}{(c_1^1 - c_3^1)(c_2^2 - c_3^2) - (c_2^1 - c_3^1)(c_1^2 - c_3^2)}$ (Eq. 5)

$f^2 = \left(\frac{(c_t^1 - c_3^1)}{(c_2^1 - c_3^1)}\right) - \left(\frac{(c_1^1 - c_3^1)}{(c_2^1 - c_3^1)}\right) f^1$ (Eq. 6)

$f_3 = 1 - f_1 - f_2$   (Eq. 7)

For the EMMA analysis, it was assumed that snow, groundwater and glaciers contributed to the Cuevas River in December. The first source to melt in December is snow, which was even observed in the samples taken in the Mt. Tolosa rock glaciers conglomerate (Fig. 11 and 13). According to field observations and the results of MODIS images, in addition to the data of

"Lagu" station, most of the snow had already melted by January and February for this particularly dry period. Thus, in January and February, it was assumed that the sources contributing to the Cuevas River changed, being dominated by rock glaciers, groundwater and glaciers.



## 3. Results

### 3.1 Streamflow and environmental variables analysis

#### 3.1.1. Horcones Inferior Glacier

Streamflow of the Horcones Inferior Glacier showed a similar variability as that of temperatures (Fig. 2a). The best overall linear model obtained for the total measured variables in both, "France" and "Lagu" stations, includes only mean daily air temperature, being significant in both stations ($p < 0.01$). The response variable behaviour (average daily streamflow delivered by the Horcones Inferior Glacier) were explained by the model: QMD = 0.2221 * TMD air (France) + 0.9243 ($R^2 =$ 0.7); and the model QMD = 0.2318 * TMD air (Lagu) - 0.6489 ($R^2 = 0.6$).

The relationship ($R^2 = 0.89$) between the average daily air temperatures measured at both, "Lagu" station (3043 m a.s.l.) and "France" station (4016 m a.s.l.) was also significant. The difference in temperature between the "France" station (973 m uppermost than "Lagu") is negative by about 7 ° C and the temperature decreases by 0.73 ° C every 100 m a.s.l., according to the equation: TMD air France = -1.03833 * TMD air Lagu - 7.08138.

#### 3.1.2 Mt. Tolosa rock glaciers conglomerate analysis

When the analysis was performed using a generalized linear model for the climatic variables from Lagu and France stations, the most influential variables in the emergent flow of the analyzed rock glacier cluster ($R^2 = 0.56$) were mean air temperature ($p = 0.009$) and mean daily maximum air temperature ($p = 0.027$), both measured in "France" station. A generalized linear modeling was performed based on this result, considering the response variable (average daily flow) with respect to the significant variables ($R^2 = 0.49$). Subsequently, an inference of models with elimination of variables according to their significance was followed to determine the relative importance of each predictor variable, in order to simplify the model until the minimum number of variables explaining the behavior of the flow rate. In this way, it was possible to infer that the most significant variable for the flow behavior is the average daily air temperature measured in "France" station.

As shown in figure 2b, the emergent flows of the analyzed rock glacier conglomerate could be adjusted through a third–order polynomial equation with mean air temperature values measured at "France" station ($y = 4E^{-5}x\ 3 - 9E^{-5}x\ 2 - 0x + 0.007$; R² = 0.62), at 4016 m a.s.l., which is the closest measurement to the approximate elevation of the rock glaciers. This could indicate a certain threshold, around 6 °C, is needed for a higher flow delivery rate, as the air temperature increases.

### 3.2 Hydrochemical analysis

#### 3.2.1 Glacier covered basins performance

The yield indicates how many times more (or less) water produces a basin per area unit regarding the average production of the basin that drains (Milana, 1998). Average water productivity per area unit was calculated according Equation 8:





R= (Q$_{gl}$/A$_{gl}$)/ (Q$_r$/A$_r$) (Eq. 8)

For:

Q: streamflow

A: area

R= yield

Where,

Q$_{gl}$-A$_{gl}$: streamflow (m$^3$ s$^{-1}$) and area (km$^2$) for the glaciated basin, respectively.

Q$_r$-A$_r$: streamflow (m$^3$ s$^{-1}$) and area (km$^2$) for the basin receiving the sub–basins drains (Cuevas River basin, in this case), respectively.

When analyzing the average water production of glaciated basins yield with respect to the production of the Cuevas River basin for the different stages of the melting period, the Horcones Inferior Glacier basin yields 307% in March to 625% in December (Fig. 3). Meanwhile, the average productivity of the Mt. Tolosa Rock Glaciers conglomerate accounts from 34 to 130% for the delivery periods of ice and snowmelt waters, with reference to the Cuevas River, respectively (Fig. 3).

### 3.2.2 Rivers ionic characterization

Reinforcing the results presented in Crespo et al. (2017), in this new set of data obtained in the (2013–2014) summer period, Cordillera Principal geological province presents compositions of calcium + magnesium (ordinates axis) and sulfates + bicarbonates (abscissas axis) near the 1: 1 line (Fig. 4a), indicating calcite, dolomite and gypsum solutions in this first order morphotectonic unit. The correlations of Na$^+$ / Cl$^-$ and Ca$^{+2}$ / SO$_4^{-2}$ (Fig. 4b and 4c) in Cordillera Principal waters indicate that halite and gypsum solutions of evaporite sequences are processes that mainly affect water chemistry of the streams.

### 3.2.3 Electrical conductivity of different water sources from Cordillera Principal

Debris covered glaciers and rock glaciers had an average conductivity of 674 and 818 μS cm$^{-1}$ respectively. Water from springs (without "Puente del Inca" geothermal waters) had the highest average electrical conductivity, with 2185 μS cm$^{-1}$, and rivers presented intermediate values of 1154 μS cm$^{-1}$. At the lower end of the conductivity values, the snow fed basins ("Los Puquios") had the lowest values (mean value: 144 μS cm$^{-1}$). The electrical conductivities of "Puente del Inca"
geothermal water samples were very stable over time and presented values of ~ 23000 μS cm$^{-1}$ (Table S6).

The electrical conductivity evolution along the melting period for different water sources analysis did not show any trend for groundwater ("Vertiente del Inca" Spring and "Puente del Inca" geothermal waters) but it did, with a positive trend, for the Horcones Inferior Glacier water, the Mt. Tolosa Rock Glaciers conglomerate waters, "Los Puquios" Stream and the main rivers of Cordillera Principal like the Cuevas River (Fig. 5).



### 3.2.4 Isotopic composition

The isotopic composition of groundwater, geothermal waters and Cordillera Principal rivers (represented by "Vertiente del Inca", "Puente del Inca" and "Cuevas River", respectively) did not show a significant temporal trend along the delivery period sampling (Fig. 6). For the snow watersheds (i.e. "Los Puquios") a slight increase in heavy isotope composition from the beginning of the melting season to the first week of January ($R^2 = 0.34$) was observed. From mid–January onwards, a more marked heavy isotope depletion ($R^2 = 0.79$) began, accompanied by an increase in salt concentration ($R^2 = 0.87$), as observed in Fig. 5. Highly disordered stable isotopes values were observed in mid–summer for the Horcones Inferior Glacier. For the Mt. Tolosa rock glaciers conglomerate, at the beginning of the melting period, enriched samples were observed, which then stabilized at lower values (Fig. 6).

In the scatter plot (Fig. 7) different water sources with different symbols are identified according to their origin. Isotope values indicating evaporated waters (located to the right of the line in Fig. 7) correspond to some of Mt. Tolosa rock glaciers conglomerate, Los Puquios and Valle Azul snow catchments, and represent fresher snow from the beginning of the melting process in spring. The isotopic composition is clearly different between the waters of the Horcones Inferior Glacier and the waters of the Mt. Tolosa rock glacier conglomerate (Fig. 7, 8 and Table S1). The waters of the snow fed watersheds such as Los Puquios and Valle Azul are more enriched than those of the Horcones Inferior Glacier, and overlap with the values of the waters disbursed by rock glaciers.

### 3.3 Statistical analysis

### 3.3.1 Principal components analysis (PCA)

The two main dimensions of the PCA for the samples are mostly associated with the electrical conductivity and the $\delta^{18}O$ values. The first dimension accounts for 37.65% of the variability, and was associated with electrical conductivity (EC), pH and ion concentration ($HCO_3^-$, $SO_4^{-2}$, $Ca^{+2}$ and $Mg^{+2}$). The second dimension explained 18.65% of the variability, and was associated with the composition of stable isotopes ($\delta^{18}O$, $\delta^2H$ and deuterium excess or "d") and $Cl^-$, $Na^+$ and $K^+$ ions. Figure 8 shows that the first dimension, related to ion concentration, clearly separates groundwater from ice bodies and snow catchments, while the second dimension separates the different ice body types (rock glaciers vs. debris covered glaciers) and the debris covered glaciers against snow catchments. The temporal evolution of different rivers along the main dimensions of the PCA (Fig. 9), does not show a clear pattern of movement along the stable isotopes axis (dimension 2), but it does in the ionic concentration dimension (dimension 1), which would mark an increase in the concentration of salts as the melt season progresses.





### 3.3.2 Generalized linear mixed effect model analysis (GLMM): water source differentiation through electrical conductivity

The mixed model linking the fixed effect factor "water source" to the electrical conductivity response indicates a difference between ice bodies and groundwater (Table 1). The intercept (debris covered glacier) is statistically different from groundwater with respect to the electrical conductivity, but it cannot be differentiated from the waters coming from rock glaciers, snow basins, total precipitation or rivers.

### 3.3.3 GLMM: water sources differentiation through isotopic signatures

The best model from the GLMM, linking the fixed effect factor "water source" to the $\delta^{18}O$ response variable, presents an Akaike weight = 1 ($R^2$:0.67). As a result, it was possible to clearly distinguish the water discharged by the Horcones Inferior Glacier from those from any other water source, highlighting the highest significance levels of the rock glaciers, summer precipitation and snow catchments ($p < 0.001$), (Table 2). When the different sources were analyzed as intercept, it was observed that it is also possible to differentiate rock glaciers from summer precipitation ($p < 0.001$), rivers ($p < 0.01$) and groundwater ($p < 0.05$). However, it is not possible to differentiate them (the rock glaciers) from the snow catchments, which is graphically observable in Fig. 7, and are slightly different from the geothermal waters of Puente del Inca ($p = 0.098$). The snow catchments could be distinguished from summer precipitations ($p < 0.001$), rivers and streams from Cordillera Principal ($p = 0.022$) and weakly from groundwater ($p = 0.052$). Groundwater could be distinguished from summer precipitation ($p < 0.001$).

### 3.4 Temporal changes in streamflow and chemistry

Even when the Horcones Inferior Glacier mean daily streamflow responds to average daily temperature variations (Fig. 2A and 10), at some point an inverse relationship is observed (the mean daily temperature falls while the flow climbs abruptly). This could be observed in the rain registered along January 16th, 17th and 18th (Table S7) which felt as light snow in the Horcones Inferior Glacier area (Fig. 10 and Table S7), registered through the MODIS satellite imagery. The daily environmental conditions recorded by the Park Rangers at "Confluencia Camp" in the Aconcagua Provincial Park, indicate that these events represent liquid and solid summer precipitations. Although they had not been registered in the "Lagu" weather station, the rainfall stations dependent on the Secretariat of Water Resources (SRH) of "Polvaredas" (2254 m a.s.l., 32.79ºS – 69.65ºW) and "Cacheuta" (1268 m a.s.l., 33.02ºS – 69.12°W), registered the event. It was also recorded by the "Punta de Vacas" station (2401 m a.s.l., 32.85°S – 69.76ºW) dependent on the National Meteorological Service (SMN).

In the intermediate dates of these observations, no precipitation was recorded for the "Punta de Vacas", "Cuevas", "Polvaredas", "Horcones" and "Cacheuta" stations, and for the MODIS imagery, analyzed during the study period.

When analyzing the stable isotopes composition of Mt. Tolosa rock glaciers conglomerate pro–glacial stream, a higher deuterium excess stability is observed. As shown in figure 13, the initial sampling presents negative deuterium excess values,



coinciding with the remaining presence of snow. When the snow was completely melted, the deuterium excess values rose close to 10, since 12/20 sampling, and then the isotopic composition remained very stable.

When the MODIS images were analyzed with respect to the digital elevation model, the lower limit of snow reaches the highest altitude of the Upper Mendoza River basin (recorded on 12/24/2013, SCA = 0.35%), followed by the very dry year 2011, the period studied in this work and in Crespo et al. (2017), respectively.

## 3.5 End member mixing analysis (EMMA)

For a mixing model of three components and two tracers, the mixing spaces are defined by two tracers ($\delta^{18}O$ and electrical conductivity in this case). When plotted, the three components should form the vertices of a triangle and all the river samples must be framed by the triangle. If this norm is not fulfilled, the tracers are not conservative, or there may be contributions from other sources not characterized, as Cuevas River in March (Figure 12).

Thus, when calculating the percentages contributed by each water source in each month of the period, the results were those observed in Table 4.

## 4. Discussion

The geological province of Cordillera Principal was defined as a study area, because it receives almost exclusively winter snowfall from the Pacific, and presents a much clearer system for finding narrow differences in chemical signals compared with the rest of the formations of the Upper Mendoza River basin. This advantage was further reinforced by a period of melting framed in a mega drought period ($CR^2$, 2015), which, due to an early melting of the winter snow, allowed to define more precisely the stable isotope composition of the different water sources (i.e. glaciers, rock glaciers, snow or groundwater). The aim of this work was to detect, by chemical tracers and flow measurements, the temporal variation within a melting season of the water supply from ice, snow and groundwater origin and its correspondence with the most influential meteorological variables. Finally, the contribution of each source was calculated.

### 4.1 Environmental controls

In most cases, climatic variables such as precipitation and temperature are the dominant influences in the formation and survival of glaciers (MacDonell et al., 2013; Karpilo, 2009). As the analyzed period includes a complete ablation season and not the accumulation period in winter, the temperature, which plays an important role in the ablation of the Central Andean glaciers, would dominate variability of water discharge (Corripio et al., 2007; Pellicciotti et al., 2008). This can be observed in Fig. 2 and 10, where temperature becomes a significant variable controlling the behavior of melt flows.

The flow rate of the Horcones Inferior Glacier fits a linear response with the daily mean air temperature. In the case of the Mt. Tolosa rock glaciers conglomerate, the delivery of the melt flow in relation to the temperature could be adjusted through a third–order polynomial equation (Fig. 2b). This could be explained via the isolation created by the debris layer, which in



turn makes a more delayed thermal inertia for the glacier ice thawing (Østrem, 1965; Buk, 2002; Trombotto and Ahumada, 2005). In this way, rock glaciers would begin to deliver more water after certain temperature threshold is reached (in this case: around 6°C of mean daily air temperature).

As discussed in detail below, it was possible to capture the change of heavy stable isotopes composition in the Tolosa rock
glaciers conglomerate streamflow, reflecting the snow to ice melt contributions. The peak of the spill is just after this transition, reflecting that it is composed of ice melt water and not of snowmelt (Fig. 11).

However, the maximum discharge of the melting from active layers, occurring at the end of December, leads to two possible situations. The first hypothesis would be linked to a loss through infiltration. This could be due to a water loss through a fractured aquifer after the spill peak, without being detected in this study.

The second hypothesis would be linked to mean daily air temperature positive trend recorded in "Lagu" station until 12/24/2013 ($R^2$=0.46). After this date, the trend becomes negative ($R^2$=0.50). The first temperature increase stage generated enough energy to melt the surface portions of the active ice layer from rock glaciers, generating the increase flow until 12/24/2013. A time span without precipitation for more than one month (from November 22[nd] to December 24[th], 2013) accompanied this period. This flows and air temperature dynamics in rock glaciers coincides with the studied in Cordón del
Plata, Cordillera Frontal geological province, by Trombotto et al. (1997). After the discharge maximum, the entrance of an eastern front is recorded on 12/25/2013 (Table S7), which abruptly decreases the temperature and thus the available energy for melting processes. From this point, a curve of water depletion begins and cannot be recovered again.

Based on the EMMA analysis and field observations (were posterior rock glaciers water contribution in other areas were observed), we believe that the first option is likely possible (i.e.: a water infiltration that could not be detected within this
work).

## 4.2 Ice, snow, and rain contributions to streamflow

The PCA analysis showed a clear distinction of debris covered glaciers with groundwater, rock glaciers and snow basins (Fig. 8). Ice bodies and snow basins ion concentration differ from those basins with groundwater predominance. The temporal evolution along the main dimensions of the PCA (Fig. 9) does not show a significant movement in stable isotopes
(dimension 2) of rivers chemical composition, while it shows an increase of ions concentration (dimension 1), as the melting season progresses, suggesting a growing contribution of groundwater.

While may not be possible to distinguish particular samples of winter snow from others of glacier melt in summer, the average of all winter snowfall through the result of the snow basins streamflow (Crespo et al., 2017), can be clearly noticed. The isotopic enrichment observed in the waters from "Los Puquios" snow basin until the first week of January (Fig. 6), could
be due to isotope elution. In this process preferential initial melting of snow with lighter stable water isotope composition is followed by isotope enrichment of water, as melting period progresses (Ohlanders et al., 2013). The subsequent impoverishment in heavy isotopes and increase in saline concentration (Fig. 5) would indicate the increasing contribution of





snowmelt from upper areas, with lower stable isotopes composition (due to higher altitude provenance) and longer sediment contact times (increasing salinity).

Samples of the Santa María stream have intermediate stable isotopes values between glacier and snow basins compositions (Fig. 7). This could be attributed to contributions of two small ice bodies detected by the national glacier inventory

(IANIGLA–ING, 2015a).

River isotopic compositions (represented by the hollow blue circle in Fig. 7) are scattered, with impoverished values similar to glaciers and enriched up to the lowest values of pure snow catchments such as "Los Puquios" or "Valle Azul". Although the rivers of the Cordillera Principal would feed on winter snow in normal to abundant snow loads years (Masiokas et al., 2006), for this dry period this stable water isotope composition places the major rivers of the main mountain range within

values more common to those of glacier composition than to those of snow basins.

The isotopic composition changes observed in the Horcones Inferior glacier streamflow, from the beginning to the end of the melting season, seems to be linked to summer storm events. According to MODIS images, since it was a very dry period, almost all accumulated winter snow had melted in this ice body basin before the sampling period (Fig. 10). Increase in the natural abundance of heavy isotopes was punctual and responds to the contributions of sporadic summer storms, much more

isotopically enriched (Fig. 6 and 10).

Figure 10 shows, following the precipitation records listed in Table S7, isotope enrichment in waters from the proglacial stream, according to the enriched waters provided by summer storms incoming from the Atlantic Ocean. Samples from January 19[th] and 22[nd] show enrichments due to summer precipitations on January 16[th] to18[th], 2014. The January 29[th] sampling shows the enrichment due to the January 26[th] and 27[th] summer storms. Then all the snow melted, followed by a

glacier water stable isotope impoverishment until 2/13/2014. Less successive snow melt contributions from 1/27/2014 storm where also detected. On 2/19/2014 another enrichment is registered due to the February 14[th] until 18[th] storm. On 2/26/2014, this enrichment is again registered for the contribution melting from light snowfalls of February 24[th] and 25[th], and then impoverishing until the next sampling (3/15/2014), to the mean values of the proglacial stream. March 1[st] and 2[nd] storms were not recorded because no samplings were conducted close to those dates.

Summarizing, whenever the Horcones Inferior Glacier stream was sampled after a storm, an increase in stable isotopes values was observed, according to the most enriched values of summer storm precipitation waters (Fig. 10).

In the case of the Mt. Tolosa rock glaciers conglomerate it was possible to access and sample since November 2013, and capture the change in the isotopic signatures of the stream during the transition from snow to ice melting. As shown in Fig. 7, the first two samplings corresponding to the first three weeks sampled (from November 13[th] to 26[th]) have been separated,

being named for this graph as "Tolosa Snow". These samples present lower deuterium excess values expressing more evaporated conditions (Fig. 11) than the other samples taken in the same site (labeled as "Tolosa Rock Glacier" in Fig. 7).

At this early date, snowfall of November 14[th]; 19[th] and 21[th], and subsequent snowmelt were recorded, according to MODIS images for the rock glaciers conglomerate area (Table S7 and Fig. 11). In this stage, the water from the proglacial stream presents very similar composition to those of the analyzed basins in the first spring sampling (Fig. 7), indicating a major




contribution of snowmelt (Fig. 11). The transition samples (Fig. 11 and blue triangles in Fig. 7), showed a transition where the snow and ice contributions would be mixed. Finally, since the 12/20/2013 sampling, streamflow stable isotopic composition was similar to that of the rock glacier ice melting (more impoverished and less evaporated).

As was explained before, this heavy stable water isotope composition, served as a basis for understanding the unimodal discharge of water that was recorded in the rock glaciers effluent stream. Summarizing, the discharge peak was determined by the delivery of glacial ice melt water instead of the presumed snow melt contribution that could be interpreted by the end of December 2013 if only MODIS imagery were considered.

It was demonstrated that the flows delivered by ice bodies respond to specific meteorological conditions (temperature or snowfall events) in a significant way and these changes can also be observed in the water supplied by the ice bodies isotopic composition variability for both: The transition from snow to ice and singular precipitations generated by convective summer storms of Atlantic origin that reach the Cordillera Principal, surpassing the Cordillera Frontal mountain range.

## 4.3 End member mixing analysis (EMMA) water source quantification

The variables that mostly explained the PCA variability, and presented the lowest variation coefficients for each of the analyzed sources, were the water isotopic composition ($\delta^{18}$O) and electrical conductivity. Three water source type contributing to Cuevas River in December were assumed for the EMMA analysis: snow, groundwater and glaciers. On December, the first component to melt will be the snow, which is even observed in Mt. Tolosa rock glacier conglomerate samples (Fig. 11). At the same time glacier melt and groundwater were considered to contribute to the Cuevas River. According to field observations, MODIS imagery analysis (Fig. 10 and 11) and "Lagu" station data, most of the snow had already melted for this particularly dry period before January 2014. Thus, in January and February, it was assumed that the water sources contribution to the Cuevas River changed, being rock glaciers, groundwater and glaciers (Fig. 12).

The Cuevas River basin covers an area of 676 km$^2$, where 14.29 km$^2$ are occupied by uncovered glaciers (equivalent to 2.11% of the total basin area), 13.21 km$^2$ by debris covered glaciers (equivalent to 1.95% of the basin area); 1.04 km$^2$ by snow patches (equivalent to 0.15%); 11.9 km$^2$ by debris covered glacier and rock glacier (equivalent to 1.76%) and 17.6 km$^2$ by rock glaciers (equivalent to 2.6% of the total basin area) (IANIGLA–ING, 2015a). According to the National Glacier Inventory classification (IANIGLA–ING, 2015a), ice bodies can be separated into two large groups: glaciers and rock glaciers. Without pure rock glaciers (uncovered glaciers, glaciaretes, debris covered & rock glaciers), 6% of the total basin area (40.44 km$^2$) is covered by glaciers.

Although glaciers represent 6% of the total area of the basin (IANIGLA–ING, 2015a), during the analyzed melting period, the EMMA analysis shows a contribution between 8 to 9% in the months of January–February, respectively, to the total Cuevas River streamflow, and up to almost 20% in December (Table 4). This high percentage calculated for December may be due to a snowmelt from the upper elevation bands for this dry cycle, which is not detected with the markers and characterized sources. However, it is clear that in January and February (when the signal can be purely attributed to ice





melt), ice bodies contribute 8.5% (averaging both months), a 142% with respect to the percentage area they occupy in the Cuevas River basin (6%).

Rock glaciers occupy 2.6% of the Cuevas River basin area, but contribute between 47 to 34% of Cuevas River flow in January and February, respectively. This represents a contribution between 1807% and 1307% to the total flow of the Cuevas River basin, regarding the area occupied by rock glaciers.

Snowmelt, considered as a contributor only in December under this analysis assumption, contributes 48% to the total flow of the Cuevas River. Groundwater contributes 32, 45 and 56% for the Cuevas River flow rates corresponding to December, January and February, respectively. This emphasizes the importance of this source, which may represent either groundwater or the continuous and discontinuous permafrost melting.

This approximation points to the periglacial and glacial environments contribution relevance, particularly during this analyzed very dry year, with ice bodies contributing most of the Cordillera Principal rivers streamflows.

## 5. Conclusions

Distinct water sources differ in composition along a melting season for both, stable water isotopes and ion chemistry. This reinforces what had been observed at a seasonal scale by Crespo et al. (2017) and allows to detect and estimate the relative contribution of snow, groundwater and ice bodies to rivers.

The air temperature strongly modulates the glaciated basins streams flow, presenting a delayed thermal inertia for rock glaciers. The convective summer precipitation events were detected by pro–glacial streams water stable isotope composition (even when such events were not detected in weather stations), allowing to pair singular flow increases due to particular summer storm events. The snow to ice contribution transition could also be detected in a sub weekly resolution.

This study contributes to better understand the hydrological behavior through the discernment of the contribution from different water sources over time. It was possible to distinguish and quantify the sources generating the flow and to improve the understanding of the most influential environmental controls in the melting and generation of water flows. By deepening our understanding about the delivery and depletion dynamics of different water sources influencing the hydrological processes of the Mendoza River basin, the vital artery of this arid and fragile territory, these tools are expected to serve to decision makers and to generate the necessary mitigation policies for an improved water resources administration along the actual and future climate change scenarios.

## Acknowledgements

To Esteban Jobbagy for his help with the isotopical analysis in the Environmental Studies Group (GEA), Applied Mathematics Institute (IMASL – Conicet). To Álvaro Gonzalez–Reyes for his suggeences. To DRNR of Mendoza province,



for the permission to deploy collectors, carry out sampling, and work with park rangers in protected areas (A. Zalazar, V. Ottero, R. Massarelli, O. Aranibar and J. Gimenez). To Alejandro Casteller, Juan Pablo Scarpa, Gustavo Costa, Mariano Castro, Laura Zalazar, Ivana Pecker and Marcelo Quiroga for their unvaluable help in fieldworks.

## Funding

This work was supported by the National Agency for Science and Technology Promotion under grant PICT 2011(2703), and National University of Cuyo under grant PID (2011–2013 and 2013–2015).

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



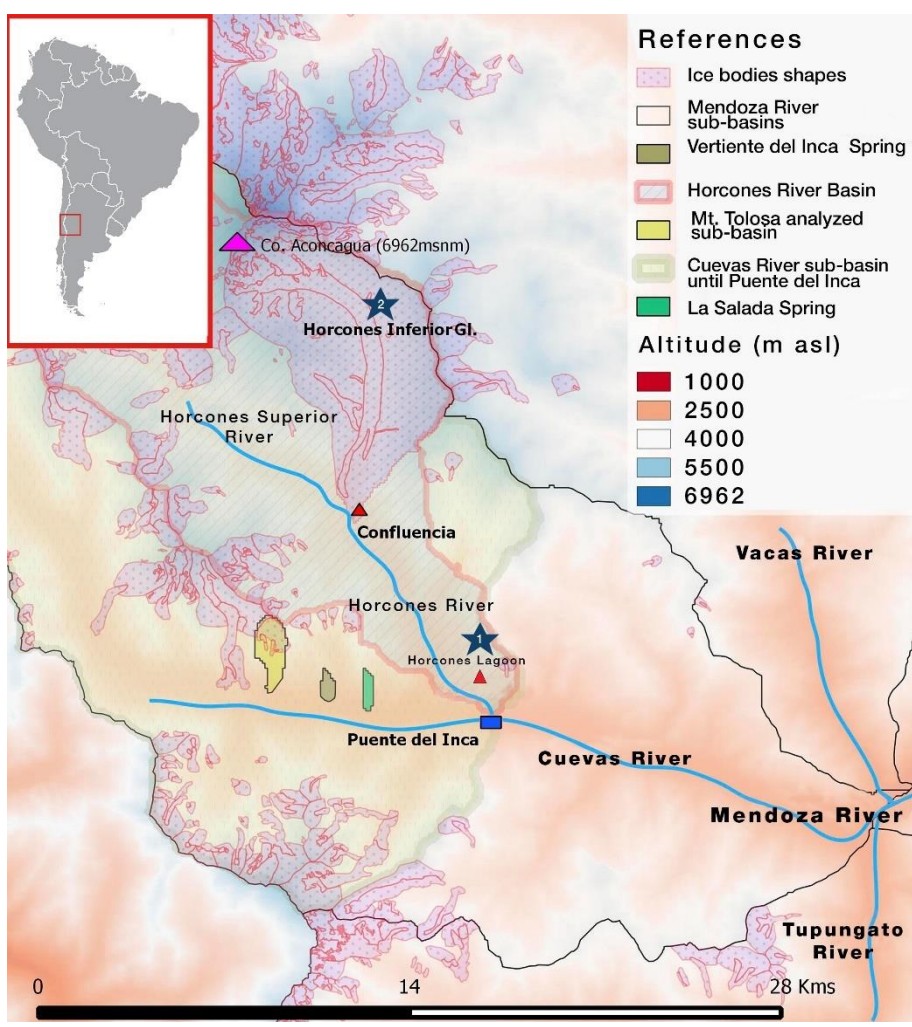

**Figure 1:** Digital elevation model map and sampling reference areas. Puente del Inca is marked with a blue rectangle. The Confluencia Camp, where the rainfall collector and the Confluencia Vieja and Nueva springs are, is marked by a red triangle. The rainfall collector Horcones Lagoon is also marked with a red triangle. The weather stations of Horcones DGI and HOBO Plaza Francia are marked with a blue star and numbers 1 and 2, respectively. The corresponding glacier shapes (marked with red contour and violet dotted edges) were taken from the glacier official inventory (IANIGLA–ING, 2015a and 2015b). Cuevas, Vacas and Tupungato rivers when join in Punta de Vacas, form the Mendoza River.The logo of Copernicus Publications.




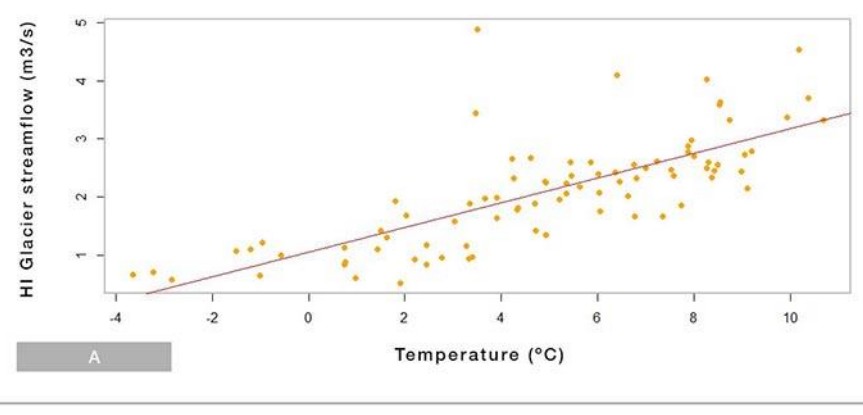

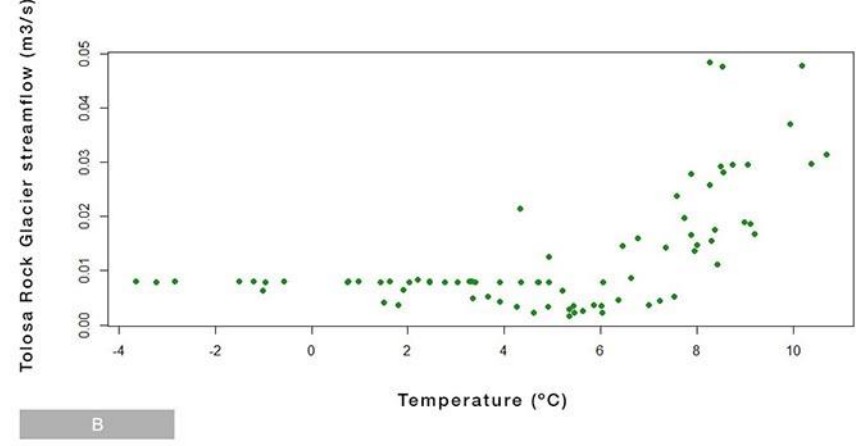

**Figure 2:** A) Scatter plot of the Horcones Inferior Glacier streamflow and mean daily air temperature. The line represents the adjustment of the model. B) Scatter plot of the average daily flow from Mt. Tolosa rock glaciers conglomerate and mean daily air temperature.



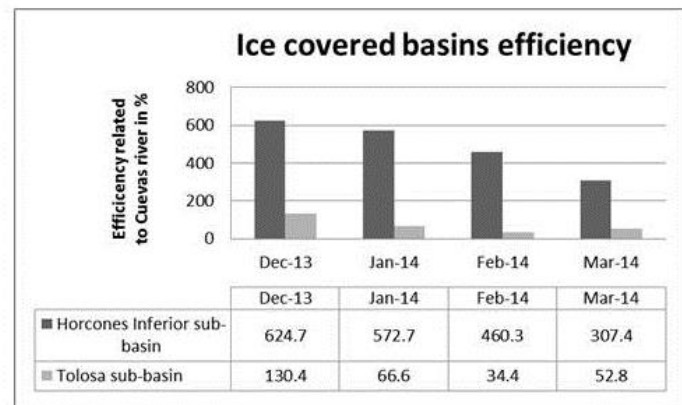

| | Dec-13 | Jan-14 | Feb-14 | Mar-14 |
|---|---|---|---|---|
| ■ Horcones Inferior sub-basin | 624.7 | 572.7 | 460.3 | 307.4 |
| ▩ Tolosa sub-basin | 130.4 | 66.6 | 34.4 | 52.8 |

**Figure 3:** Mean water yield for glaciated basins (Horcones Inferior sub–basin) and rock glaciers (Tolosa sub–basin) respect to the Cuevas River mean water production, which receives the drains from the sub–basins in different period of snowmelt contribution (December 2013) and ice melt contribution (Jan–Mar 2014).





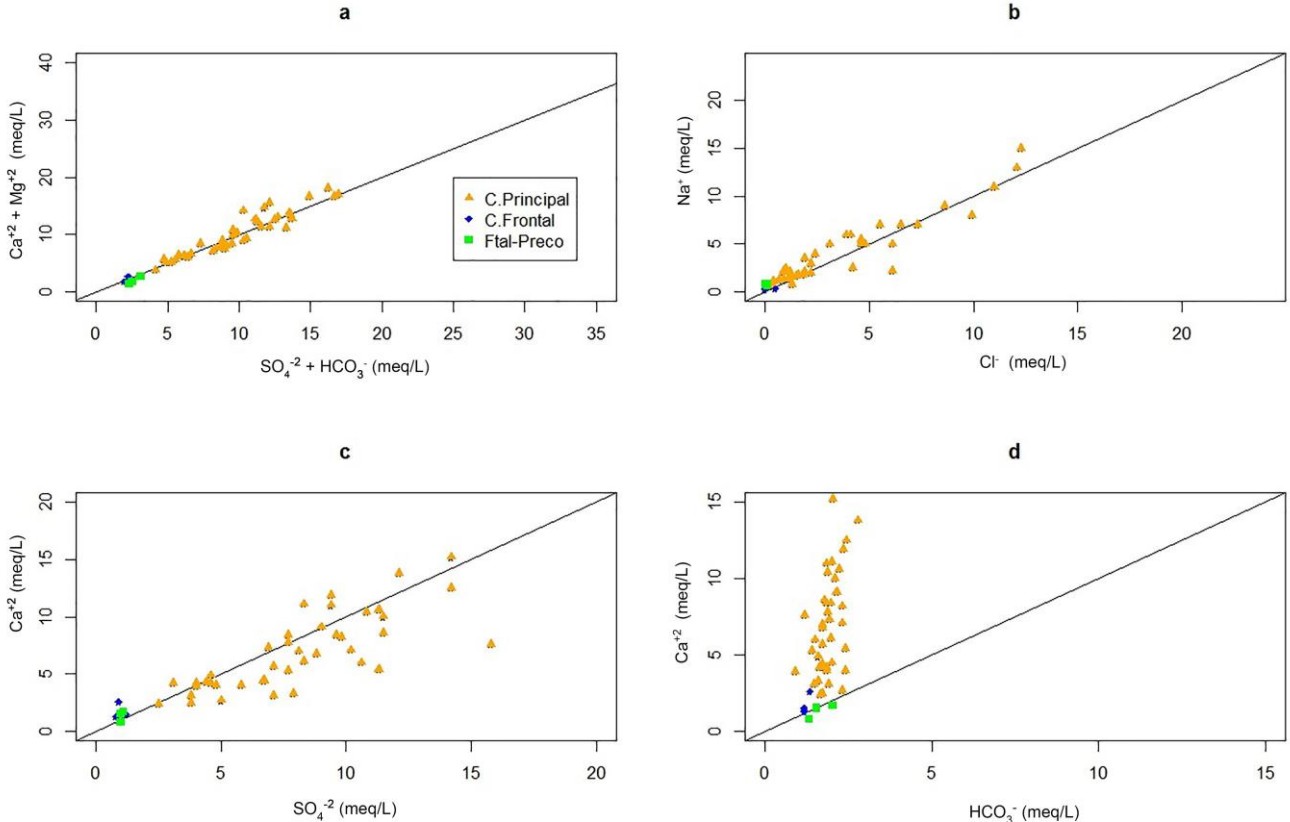

**Figure 4:** Scatter plots of: a) $Ca^{+2} + Mg^{+2}$ and $SO_4^{-2} + HCO_3^-$, b) $Na^+$ and $Cl^-$, c) $Ca^{+2}$ and $SO_4^{-2}$ and d) $Ca^{+2}$ and $HCO_3^-$, explaining dissolution processes. The results of samples belonging to the waters of a river in the Cordillera Frontal (Colorado River) and a mixture of waters from Cordillera Frontal with Precordillera (Ftal–Preco) additionally sampled from the Uspallata Stream are presented as a reference in this graph.




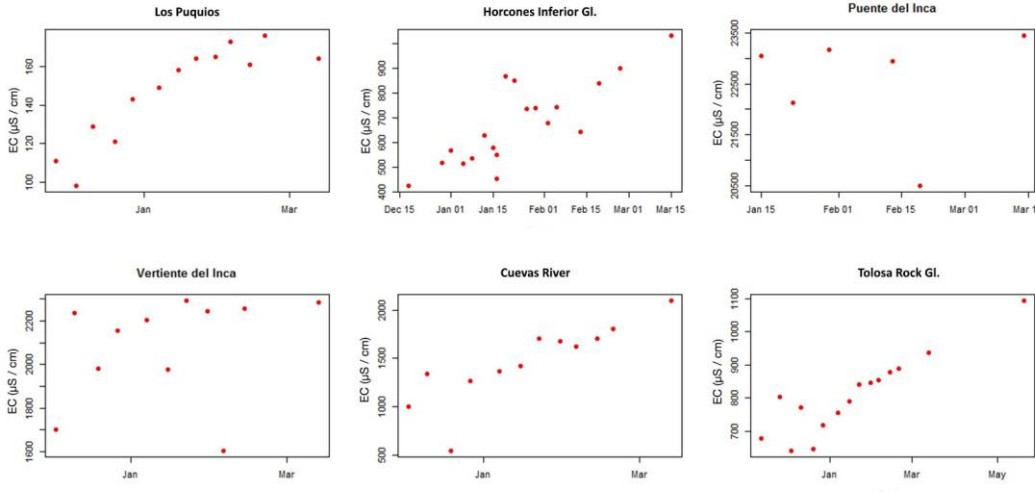

**Figure 5:** Water sources electrical conductivity along the melting season. "Los Puquios" represents the snow dominated basins, "Vertiente del Inca" represents groundwater and "Puente del Inca" geothermal waters.

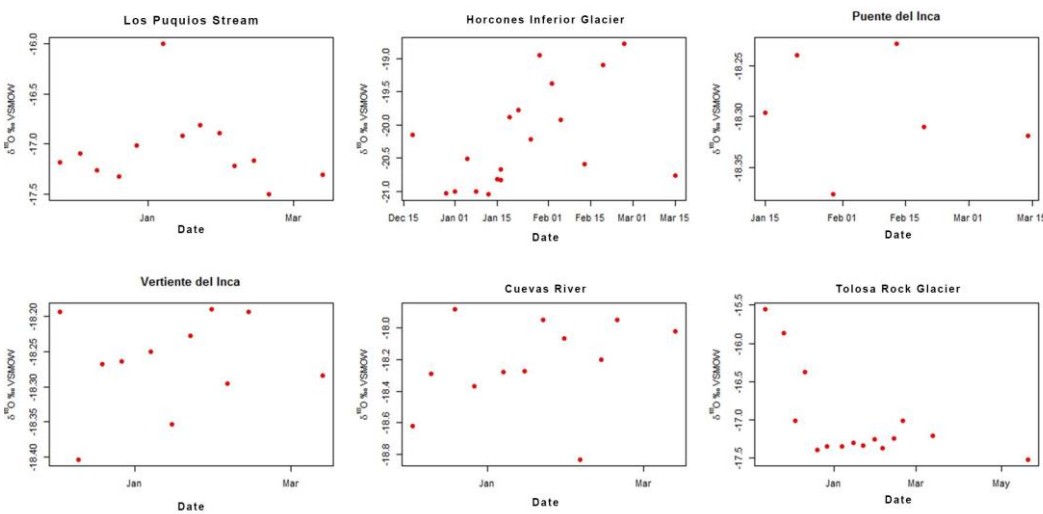

**Figure 6:** Different water sources $\delta^{18}$O temporal evolution.





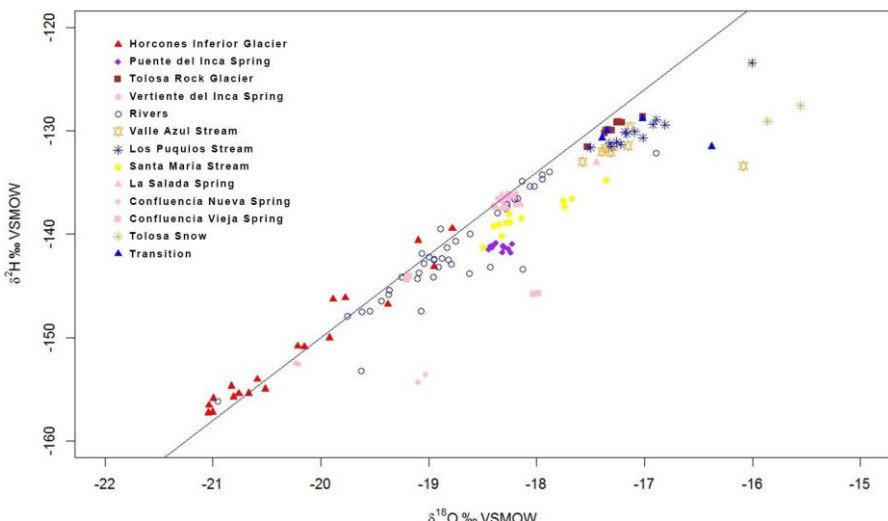

**Figure 7:** $\delta^{18}$O and $\delta^2$H values scatter plot of the analyzed samples. The adjusted line is the global meteoric water line (Craig, 1961). Samples from the Mt. Tolosa rock glaciers conglomerate showing a mixture of ice and snow signatures were marked as "Transition". Summer precipitation samples from "Lagu" and "Confluencia" are not represented in the graph as they are out of scale.

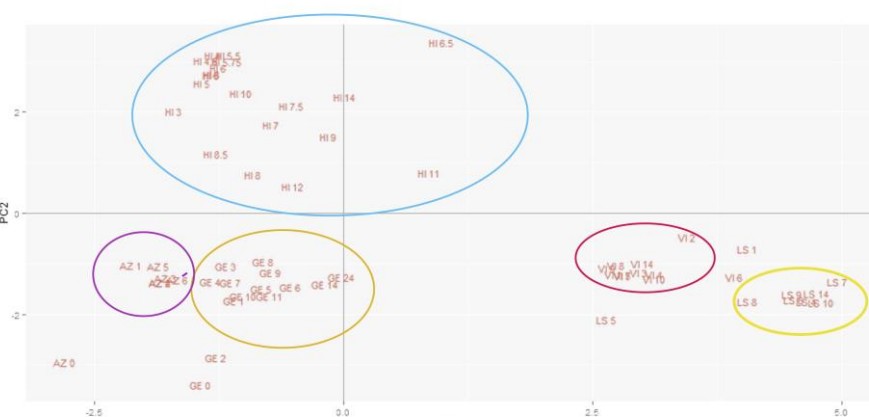

**Figure 8:** Temporal evolution of samples from: Horcones Inferior Glacier (HI), Mt. Tolosa rock glaciers conglomerate (GE) and Valle Azul snow catchment (AZ). The Vertiente del Inca (VI) and La Salada (LS) streams represents the groundwater samples. The numbers followed by the acronyms mark temporality for weeks, beginning at week 0 through 24. The fractions show samplings between weeks.





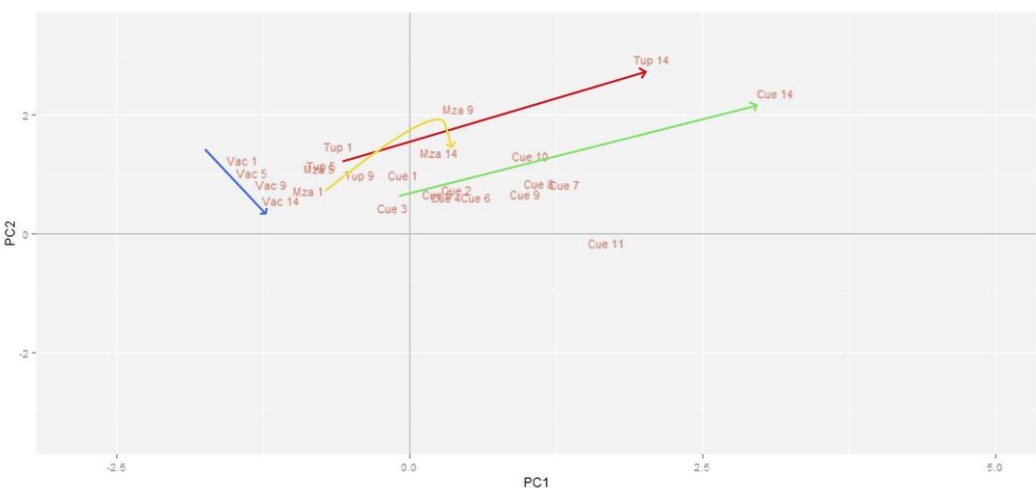

**Figure 9:** Temporal evolution of the PCA for the main rivers of the Cordillera Principal geological province sampled in Punta de Vacas. Vacas River (Vac); Tupungato River (Tup) and Cuevas River (Cue) form the Mendoza River (Mza) when they join in Punta de Vacas. The arrows mark the temporal evolution through the melting season, beginning in week 0 until the week 14[th].

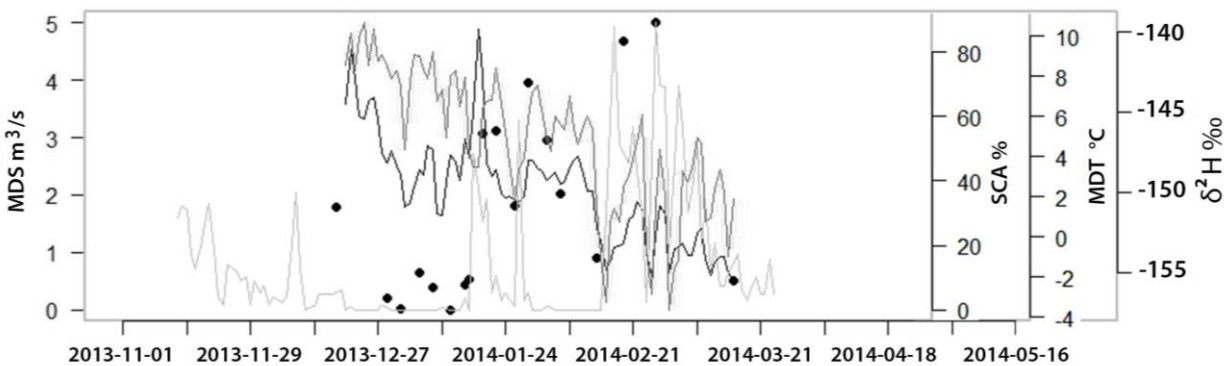

**Figure 10:** Snow covered area (SCA), streamflow (MDS), mean daily air temperature (MDT) and deuterium composition of the Horcones Inferior Glacier. The black line shows the MDS, the light grey line the SCA and the dark grey line MDT of "Francia" weather station. The black points are the $\delta^2$H water composition.





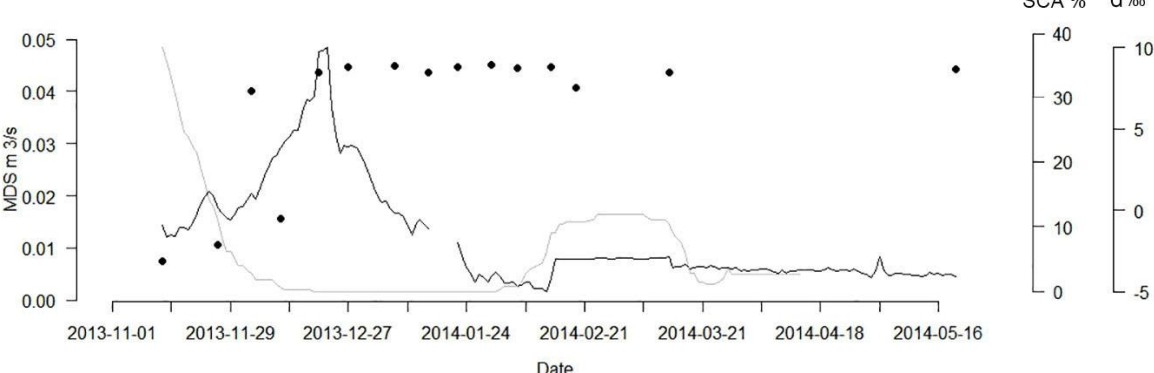

**Figure 11:** Mean daily streamflow (MDS) of Mt. Tolosa rock glaciers conglomerate (black line) and snow covered area percentage (SCA, grey line) as a moving average with 31–days bilateral windows obtained with the MOD–MYD product. The dots are the deuterium excess values (d‰).

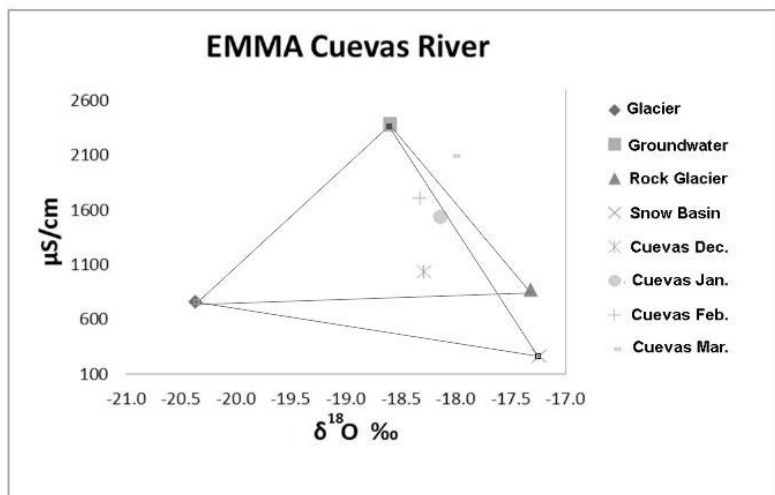

**Figure 12** Dispersion plot of mean stable water isotope composition and electrical conductivity of different water sources draining to the Cuevas River. The values per each water source and Cuevas River per month are expressed in Table 3.



**Table 1:** Mean, standard deviation and confidence interval for different water sources in comparison with the intercept (debris covered glacier). The electrical conductivities significant differences between debris covered glaciers (intercept) and the rest of the water sources are reported. The significant codes are: 0; '***' 0.001; '**' 0.01; '*' 0.05; '.' 0.1. The significant variables are marked in bold.

| Variable | Mean | SD | CI (2.5–97.5%) | p | |
|---|---|---|---|---|---|
| Intercept | 674.1 | 458.9 | -225.41 & 1573.62 | 0.14 | |
| Summer precipitation | -289.1 | 775.1 | -1808.22 & 1230.01 | 0.71 | |
| Rock glacier | 144.3 | 742.2 | -1310.32 & 1598.97 | 0.85 | |
| River | 447.6 | 561.7 | -653.23 & 1548.49 | 0.43 | |
| Valle Azul snow basin | -243.5 | 744.1 | -1701.92 & 1214.96 | 0.74 | |
| Los Puquios snow basin | -527 | 742.4 | -1982.06 & 928.02 | 0.48 | |
| Sta. María basin | -440.3 | 742.6 | -1895.76 & 1015.22 | 0.55 | |
| Groundwater | 1495 | 587.6 | 343.29 & 2646.65 | 0.01 | * |

**Table 2:** Mean, standard deviation and confidence interval for water source and the intercept (debris covered glacier). The $\delta^{18}O$ statistical differences between debris covered glaciers and the rest of the water sources are reported. The significant codes are: 0; '***' 0.001; '**' 0.01; '*' 0.05; '.' 0.1. The significant variables are marked in bold.

| Variable | Mean | SD | CI (2.5–97.5%) | p | Signific. |
|---|---|---|---|---|---|
| Intercept | -20.23 | 0.56 | -21.33-19.13 | $<2e^{-16}$ | |
| Puente del Inca | 1.89 | 0.79 | 0.34 & 3.45 | 0.0168 | * |
| Summer precipitation | 7.63 | 0.92 | 8.83 & 9.43 | $<2e^{-16}$ | *** |
| Rock Glacier | 3.22 | 0.89 | 1.48 & 4.96 | 0.0003 | *** |
| Rivers and streams | 1.43 | 0.68 | 0.09 & 2.76 | 0.0353 | * |
| Valle Azul snow basin | 3.07 | 0.91 | 1.28 & 4.86 | 0.0008 | *** |
| Los Puquios snow basin | 3.18 | 0.89 | 1.43 & 4.93 | 0.0004 | *** |
| Santa María basin | 2.15 | 0.89 | 0.39 & 3.90 | 0.0165 | * |
| Groundwater | 1.61 | 0.71 | 0.21 & 3.01 | 0.0238 | * |



**Table 3:** Water sources and Cuevas River mean stable water isotope composition and electrical conductivity values

| Component/Tracer | $\delta^{18}O$ ‰ | CE µS/cm |
|---|---|---|
| Glacier | -20.4 | 762 |
| Groundwater | -18.6 | 2382 |
| Rock glacier | -17.3 | 873 |
| Snow basin | -17.2 | 265 |
| Cuevas River December | -18.3 | 1037 |
| Cuevas River January | -18.1 | 1540 |
| Cuevas River February | -18.3 | 1710 |
| Cuevas River March | -18.02 | 2094 |

5    **Table 4:** Percentage contribution from different water sources to the Cuevas River since December 2013 to February 2014. The null values from snow contribution and rock glacier aren´t inferred from this calculus are assumed from the satellite imagery analysis, field observations and chemical characterization from the rock glacier composition.

| River | Water Source/rate per month | Dec | Jan | Feb |
|---|---|---|---|---|
| | Glacier | 19.7 | 8 | 9 |
| Cuevas | Groundwater | 32 | 45 | 56 |
| | Rock Glacier | 0 | 47 | 34 |
| | Snow | 48 | 0 | 0 |