# Peer review of "Mountain water cellars: a chemical characterization and quantification of the hydrological processes and contributions from snow, glaciers and groundwater to the Upper Mendoza River basin (~ $32 \circ S$ ), Argentina"

_Hydrology and Earth System Sciences, 2018_

## Referee Comment (RC1) · Anonymous Referee #1 · 20 Jun 2018

The manuscript "Mountain water cellars: a chemical characterization and quantification of the hydrological processes and contributions from snow, glaciers and groundwater to the Upper Mendoza River basin (~ 32 ° S), Argentina" reports on different hydrological and chemical analysis of data collected in the Cordillera Principal. These analysis cover streamflow, environmental variables, hydrochemical properties, electrical conductivity, isotopic composition and other statistical analysis, such as Principal Component analysis, the application of a Generalized linear mixed effect model or an end member mixing analysis. The analysis performed in the manuscript is based on data generated in a comprehensive measurement campaign covering about 5 to 6 months. Generally, the content can be of interest for potential HESS-readers. However, there are substantial deficits, which must be addressed by the authors:

(1) An English proofreading must be performed for the manuscript, also including the figure captures. It is out of scope of this review to address the frequent grammatical deficits or the necessity to rephrase sentences (e.g. "Cuevas, Vacas and Tupungato rivers when join in Punta de Vacas, form the Mendoza River." (P22L7-8) should be "Cuevas, Vacas and Tupungato rivers form the Mendoza River in Punta de Vacas". It is unclear, what "The logo of Copernicus Publications" means in this figure capture.). There are many examples which could be provided here.

(2) The manuscript is generally quite long and it would be helpful to streamline the text and to leave out parts which are not essential.

(3) Many measurements were performed and are used in the analysis presented in the manuscript. It would therefore be good to add a section "Study Area and Data Basis" to the manuscript. Here, the authors should add a table summarizing all the measurements performed, which will help the reader to keep an overview. Additionally, the general hydrological characteristics and setting should be described (i.e. long-term mean values of precipitation, discharge, evapotranspiration, temperature etc.). After this overview, it is easier to describe the methods applied, without having to refer to settings of the measurements (as is the case now).

(4) All locations and regions mentioned in the manuscript, figures and figure captions should be consistent (what is currently not the case, e.g. in Fig. 5 and 6; in Fig. 4, for the first time, the Colorado river and Uspallata Stream are mentioned, without further references in the text).

(5) In Fig. 1 an overview map is given. Here, substantial improvements are necessary, also in the context of giving the reader a better overview. In the map, measurement locations presented in the table (see (3)) should be displayed. From the map, the topography is not easily understandable – adding a hillshade layer and using different elevation colours would help in this context. The colours and symbology used should be adopted for easier readability (e.g. The boundary colour of the Horocones River Basin is basically the same as the ice bodies). "References" should be "Legend". Other relevant information and locations, which are mentioned in the text, should be included in the map (e.g. Punta de Vacas).

(6) The quality of the figures needs to be improved. The font sizes are frequently too small and cannot be read easily. The x-axis labels in Fig. 5 & 6 should be consistent. Adding vertical grid lines, e.g. at every month, would help to analyse the temporal dynamics (e.g. begin of snow melt, glacier melt) described in the text. Using colours in Fig. 10, 11 and 12 would be helpful. Also adding a legend in the Fig. 10 and 11 is necessary.

(7) Figure 3: It is not clear, what is meant with "Ice covered basin efficiency" and "Efficiency related to Cuevas river in %".

(8) Tab. 1 & 2: No significant variables are marked in bold.

(9) Tab. 4: Table caption is one of the examples where a rephrasing is needed. The term "rate per month" is not appropriate.

(10)   Methodology: Are all sub-sections necessary?

(11)   The term for "Height-discharge calibration curves" in Hydrology is "rating curves". The authors present on P4L25 and P5L3 equations for Mt. Tolosa rock glacier and the Horcones Inferior Glacier (in the text it also called Horcones Inferior debris covered glacier). The equations should be consistently formatted as shown in Eq. 1. In general, I think that these equations do not have an added values for the reader.

(12)   P4L26-27: "The determination coefficient between calculated and observed streamflow was 0.95, for y = 0.999x - 2E-05". It is unclear what y = 0.999x - 2E-05 means in this context.

(13)   P5L4: For what is this equation? What is the difference between measured flow (Qm) and calculated flow (Qcalc), since Qm is also calculated as a function of h?

(14)   P6L3-6: Was the data of these totalizer used in the study?

(15)     P9L6-7: The equations should be formatted as shown in Eq. 1. The naming of the variables should be improved, e.g. TMD air (France) is an odd naming for a variable.

(16)     Section 3.1: What other variables were analysed apart from air temperature? I am surprised that the authors do not mention any relationship between glacier melt and solar radiation.

(17)     Section 3.2.1: What does "Glacier covered basins performance" mean? Does the mentioned Cuevas River Basin here refer to Punta de Vacas? Some interesting results, e.g. what is the percentage (not weighted by area) of the contribution of the glacier to the total runoff, is not given here, but can only be found in the discussion. Maybe it would also make sense to merge the results and discussion part.

(18)     Section: 3.2.2.: The naming of the section needs rephrasing. For what is Fig. 4d, since it is not referred to in the text?

(19)     Section 3.4: Fig. 13 is not in the manuscript.

(20)     It would be good to change the "Conclusions" into "Summary and Conclusion" and to add here the main findings and quantitative results from the manuscript, e.g. the contributions of the different water sources to the total flow. These numbers are also missing in the abstract.

(21)     In section 5, the authors write "By deepening our understanding about the delivery and depletion dynamics of different water sources influencing the hydrological processes of the Mendoza River basin, the vital artery of this arid and fragile territory, these tools are expected to serve to decision makers and to generate the necessary mitigation policies for an improved water resources administration along the actual and future climate change scenarios." (By the way - Word is giving me a warning: "Long sentence (consider revising)"). It is unclear, what is meant with "tools" in this context. What would be an example, how decision makers can generate the necessary mitigation policies for an improved water resources administration along the actual and future climate change scenarios from the results? Are these not empty phrases, which the world anyway has too many?

---

## Referee Comment (RC2) · Anonymous Referee #2 · 3 Jul 2018

General comments

This study aims to investigate the contribution of snow, glacier and groundwater to streamflow runoff generation in data sparse region Andes region. The analysis is based on meteorological and hydrological (streamflow and isotope) measurements at four stations in the period 2013/2014. The authors conclude that the contribution from glaciers increases towards summer season.

[Figure]

Overall the topic of understanding of different pathways and contribution of snow melt and groundwater processes on streamflow generation is very interesting and within the scope of the journal. This is particularly interesting in the Andes, which is the region with distinct seasonality of runoff generation processes compared to many other places in the world. However the manuscript is, unfortunately, not ready for publication in its current form. The main reasons for such critical conclusion are:

1) The significant lack of clarity of presentation, particularly the English. There are many parts which are not clearly formulated so a proof-reading by native speaker is strongly recommended.

2) The context for the analysis is not clear. Is the aim to analyse and compare the runoff generation processes during the extra drought event (2010-2015) only? If so, then I missed some more information on how this period differs/compares with a normal situation. Is this drought defined in terms of precipitation deficit only? Or also in terms of streamflow? What are the differences to other studies on such topic? Why it is interesting/important to look at it in the Andes?

3) The objectives needs to be reformulated in order to more clearly show the scientific novelty and significance compared to existing studies. The research hypotheses are in its current form rather obvious. E.g. Which environmental variables control the initial thawing. Is it not the physics and energy balance which is controlling that? I would suggest to bring forward more the context of comparative hydrology to justify the significance and contribution of the paper.

4) The data description is not rigorous. I missed more information about the temporal resolution of the data and time period. Is only one season available? Is it enough to draw some more general interpretations? Are there some other/longer data sets available?

5) I missed some more process based interpretation of the results. The linear regression between streamflow and some climatic data seems to me not enough to justify

the interpretations about the contributions of individual variables. Why not to use a hydrological model for the analysis?

6) The discussion of the results can be improved. What has been learned compared to other existing studies (i.e. related to the assessment of drought controls in other regions/climates, or related to normal situation in similar regions?) In its current form it reads more as a summary.
* * *

---

## Author Comment (AC1) · 5 Oct 2018

Responses to Referee 2, identiffied as follows: (1) comments from Referee, (2) author's response, (3) author's changes in manuscript.

Answer to Referee comment 1.

(1) The significant lack of clarity of presentation, particularly the English. There are

many parts which are not clearly formulated so a proof-reading by native speaker is strongly recommended.

(2) The manuscript was streamlined, discarding any non essential parts. We clarify the presentation, objectives and after-coming analysis. The entire manuscript is being reviewed and corrected by an ad-hoc specialist and native English speaker.

Answer to Referee comment 2.

(1) The context for the analysis is not clear. Is the aim to analyse and compare the runoff generation processes during the extra drought event (2010-2015) only? If so, then I missed some more information on how this period differs/compares with a normal situation. Is this drought defined in terms of precipitation deficit only? Or also in terms of streamflow? What are the differences to other studies on such topic? Why it is interesting/important to look at it in the Andes?

(2) We believed it is important to look at glacier-snow-river dynamics in the Andes, because many people live from water originated in these mountains. In the Central Andes more than 12 million people depend on this resource for domestic consumption, irrigation, industries, hydroelectric generation and aquifer recharge. Studies in other areas may not be extrapolated to the Andes. In addition, global change is affecting glaciers all over the world, and an understanding of glacier and river dynamics in a region with the highest peak from America contributes to the global understanding of global change. The aim of the study was clarified, and we hope it is easier to understand. Basically, the aim of the study is to estimate the contribution of different water sources to a river basin of major importance for the development of western Argentina. In particular, estimating the contributions of glaciers and groundwater justify conservation efforts being done in our and other countries, to protect glaciers and other strategic water resources for future generations. Currently, glacial and periglacial environments are protected by law, but it is not clear for policy makers and society the hydrological contribution from these environments. This study shows the importance of groundwater and rock glaciers (the

most representative periglacial environment crioform) on the water provision during dry years, when agriculture, industry and society in general suffer from water scarcity. For more clarity about the analysis period description, a figure (Fig. S1) and two tables (S8 and S9), describing the precipitation and streamflow context of the analyzed year, were added in the Supplementary data file. The context for the analysis is a melting period (December 2013 to March 2014), which in turn felt in an arid period framed by the Mega-drought. This arid condition was useful to separate more clearly the different water sources signals, without a prolonged snowmelt noise. The mega-drought is fully documented (and cited in this work) and it is not the aim of this study to describe it in the main manuscript. Because the aim of this work was to separate and quantify all the different water sources draining the basin, it should be carried out during a melting season. This works continues the work of Crespo et al. (2016), were a 2-year isotopic characterization was done. In that work a seasonal sampling time resolution was used. During that study, we started to identify some different water sources contributions. Therefore, a deeper in time resolution and focus in the main streamflow period (the melting season, which accounts for half of the year streamflow) was needed. For that purpose, we carried out this work, with a weekly resolution (instead of seasonal) along a melting period.

(3) See Figure S1 and Tables S8 and S9 in Supplementary Data file.

Context: Part of the Introduction section:

Globally, glaciers are melting at unprecedented rates and in the Andes of South America they display a widespread retreat (Masiokas et al., 2016). The water supply for the oasis irrigated by the Mendoza River depends on the recharge of snow and ice, which has been under extraordinary pressure following the mega–drought that affected the Central Andes during the 2010–2015 period (CR2, 2015; Cornwell et al., 2016). This basin is supplied mainly from snow contributions in years of normal to abundant loads (Masiokas et al., 2006). In very dry years, hydrographs show a displacement of the maximum monthly flows from January to February (Boninsegna, 2013; Lascano and

Villalba, 2007), which indicate higher contributions of glaciers to the Mendoza River flow than during average or wet years (Bruniard, 1994). According to the monthly distribution of the hydrograph, the regime of the Mendoza River could be classified in normal years as "mitigated glacial" (Bruniard, 1994, Lascano and Villalba, 2007). However, for years of extreme drought (p.e. 1968, 2010), that maximum would be transferred to the month of February (Fig. S1, Tables S8 and S9), becoming an "ultra-glacial" regime. Some authors (Leiva, 1999; Boninsegna and Villalba, 2006; Masiokas et al, 2010; Boninsegna, 2014; Lauro et al., 2016), mention that in dry years, the decreasing stream flows do not follow the marked decrease of snowfall, which would be explained by a proportionally greater contribution of ice bodies. The importance of the glacial contribution to the flows in dry years has been recognized in the region, so a national law (Law 26639) was sanctioned to map, monitor and protect glaciers as strategic water reserves. The objetives of the Argentinean National Glaciers Inventory (IANIGLA-ING, 2010), as part of the mentioned law, include the quantification of glaciers contributions to river flows, but they have not been quantified for the Mendoza River basin to date. The more widely used techniques for ice melt quantification in a glaciated basin (outflow measurements, mass balance or/and satellite images) do not allow for the quantification of temporal changes of the contributions from different sources (i.e. from snow, glaciers, rock glaciers or groundwater). Naturally occurring tracers, such as ions and isotope composition, may facilitate such differentiation. Contact time of water with air and sediments is different for water sources such as groundwater, snow, rock glaciers and glaciers, and results in distinct ions and stable isotope composition for each water source (Crespo et al., 2016). These chemical properties provide natural tracers of flow inputs along the melting season by different water sources and sub–basins to a river. In other similar glaciated basins from different geographical regions like the Bhagirathi River in Indian Himalayas (Lambs, 2000), the use of electrical conductivity and $\delta 18O$ composition served as tracers to identify water from ice, snow, and rain water. Comparable findings were published by Lambs et al. (2010) for the Garonne Valley (France), where runoff water from high altitudes was

identified using stable oxygen isotopes and conductivity data from river water samples. Similarly, Pu et al. (2013), using only $\delta$18O composition from different water sources in the Baishui River catchment (China), made a hydrograph temporal separation between rain and melt water contributions to the river, but did not differentiate between snow and ice melting (neither groundwater) contributions separately. Moreover, quantitative estimation of the diverse water sources has been achieved for the Tarim River, Central Asia, by Fan et al. (2016), where a marked seasonal variability was identified by the use of water stable isotopes and electrical conductivity. The aim of this work was to quantify the different water sources inputs from groundwater, glacial and periglacial environments along the melting season, which represents the major water contribution period in the year using two approaches: gauging glacier flows, and using naturally occurring chemical tracers.

Answer to Referee comment 3.

(1) The objectives needs to be reformulated in order to more clearly show the scientific novelty and significance compared to existing studies. The research hypotheses are in its current form rather obvious. E.g. Which environmental variables control the initial thawing. Is it not the physics and energy balance which is controlling that? I would suggest to bring forward more the context of comparative hydrology to justify the significance and contribution of the paper.

(2) The objectives were reformulated as recommended. The research novelty was more clearly expressed. The comparative hydrology analysis was incorporated (see point 3 in comment 6).

(3) Part of the new introduction section:

The importance of the glacial contribution to the flows in dry years has been recognized in the region, so a national law (Law 26639) was sanctioned to map, monitor and protect glaciers as strategic water reserves. The objectives of the Argentinean National Glaciers Inventory (IANIGLA-ING, 2010), as part of the mentioned law, include

the quantification of glaciers contributions to river flows, but they have not been quantified to date. The more widely used techniques for ice melt quantification in a glaciated basin (outflow measurements, mass balance or/and satellite images) do not allow for the quantification of temporal changes of the contributions from different sources (i.e. from snow, glaciers, rock glaciers or groundwater). Naturally occurring tracers, such as ions and isotope composition, may facilitate such differentiation. Contact time of water with air and sediments is different for water sources such as groundwater, snow, rock glaciers and glaciers, and results in distinct ions and stable isotope composition for each water source (Crespo et al., 2016). These chemical properties provide natural tracers of flow inputs along the melting season by different water sources and sub–basins to a river. In other similar glaciated basins from different geographical regions like the Bhagirathi River in Indian Himalayas (Lambs, 2000), the use of electrical conductivity and $\delta18O$ composition served as tracers to identify water from ice, snow, and rain water. Comparable findings were published by Lambs et al. (2010) for the Garonne Valley (France), where runoff water from high altitudes was identified using stable oxygen isotopes and conductivity data from river water samples. Similarly, Pu et al. (2013), using only $\delta18O$ composition from different water sources in the Baishui River catchment (China), made a hydrograph temporal separation between rain and melt water contributions to the river, but did not differentiate between snow and ice melting (neither groundwater) contributions separately. Moreover, quantitative estimation of the diverse water sources has been achieved for the Tarim River, Central Asia, by Fan et al. (2016), where a marked seasonal variability was identified by the use of water stable isotopes and electrical conductivity. The aim of this work was to quantify the different water sources inputs from groundwater, glacial and periglacial environments along the melting season, which represents the major water contribution period in the year, using two approaches: gauging glacier flows, and using naturally occurring chemical tracers.

Answer to Referee comment 4.

(1) The data description is not rigorous. I missed more information about the temporal resolution of the data and time period. Is only one season available? Is it enough to draw some more general interpretations? Are there some other/longer data sets available?

(2) The data description was better explained. This work follows the work published in Crespo et al. (2016). The 2016 paper reflects the inter-seasonal and regional analysis in a more extensive time and space resolution. The aim of this work was to characterize and quantify in a more precise time resolution (intra-seasonal, among the melting period-season) what we couldn't define at the seasonal scale. This melting period was characterized because hydrologically it is the most relevant period for this basin, accounting for ~50% of the year streamflow (Table S9). There are no other data sets available, no previous works quantifying the water sources (snow, glacial, periglacial or groundwater) contributions to the Mendoza River basin has been presently carried out. Long term streamflow data (1957-2017) and winter snow water equivalent (1987-2015) was added in the Supplementary document (Fig. S1 and Tables S8 and S9).

(3) See point 2 in Referee comment 2, Figure S1 and Tables S8 and S9 in Supplementary Data file.

Answer to Referee comment 5.

(1) I missed some more process based interpretation of the results. The linear regression between streamflow and some climatic data seems to me not enough to justify the interpretations about the contributions of individual variables. Why not to use a hydrological model for the analysis?

(2) The aim of the work was better explained in the actual version. Basically it is to quantify the different water sources contributions with chemical tracers and compare with streamflow data (in the ice bodies case). The environmental variables influences are a complement for the main analysis. It would be ideal to carry out a more complex hydrological modeling, but it would make the manuscript very extense and out of the

main scope of the present work. In fact, it has been suggested (by other referee and co-authors), to shorten this description. In the new version this section is significantly reduced. In any case, the influence of environmental variables on the streamflow was modeled with the generalized linear effects model, with proven and significant probabilities. The generalized linear effect model was carried out for all the environmental variables measured in stations 1 and 2 (Table S2). We also expected solar radiation statistical significance. Solar radiation may influence the ice melting but, during the period studied in the austral summer, the effect was not significant. The temperature (which probably includes the solar radiation effect), was significant. The results for the Horcones Inferior Glacier were expressed in section 3.1.1. The results for the Tolosa rock glaciers conglomerate are expressed in 3.1.2. (see 3).

(3) Section 2.1 reference: The Irrigation General Department of Mendoza weather station (labelled as 1 in Fig. 1), located at 3043 m a.s.l. in "Laguna de Horcones" (32.80°S – 69.95°W), measured: air temperature, soil temperature, wind speed and direction, relative humidity, incident radiation and snow water equivalent, hereinafter "station 1". Air and soil temperature HOBO sensors were also installed in the Horcones Inferior Glacier (labelled as 2 in Fig. 1), at 4016 m a.s.l. (32.69°S – 69.97°W), hereinafter "station 2" (Table S2). Both stations covered an altitude gradient of 973 m. Generalized linear models (nlme package, Pinheiro et al., 2013) in the R program (R Core Team, 2013) were conducted. The streamflow was considered the response variable, as a function of environmental data (predictor variables).

Section 3.1.1. Horcones Inferior Glacier

Streamflow of the Horcones Inferior Glacier showed a similar variability as that of temperatures (Fig. 2a). The best overall linear model obtained for the total measured variables in both, stations 2 and 1 (Table S2), includes as significant variable only mean daily air temperature ($p < 0.01$). The response variable Horcones Inferior Glacier average daily streamflow, fits linearly with mean daily air temperature for both stations, following equations 8 ($R^2 = 0.7$) and 9 ($R^2 = 0.6$), respectively:

MDS HI= 0.2221 * MDAT2 + 0.9243 (Eq. 8) MDS HI= 0.2318 * MDAT1 - 0.6489 (Eq. 9)

For: MDS: Horcones Inferior Glacier mean daily streamflow (m3 s-1) MDAT2: mean daily air temperature (°C) in station 2 MDAT1: mean daily air temperature (°C) in station 1

The relationship between the average daily air temperatures measured at both stations 1 (3043 m a.s.l.) and 2 (4016 m a.s.l.) was also significant. The difference in temperature between the station 2 (973 m uppermost than station 1) is negative by about 7 ° C and the temperature decreases by 0.73 ° C every 100 m a.s.l., according to Eq. 10 (R2 = 0.89):

MDAT2 = -1.03833 * MDAT1 - 7.08138 (Eq. 10)

Section 3.1.2 Mt. Tolosa rock glaciers conglomerate

From all the environmental variables measured in stations 1 and 2 (Table S2), the most influential variables in the emergent flow of the analyzed rock glacier cluster, were mean daily air temperature (p < 0.01) and mean daily maximum air temperature (p = 0.027), both corresponding to station 2 (R2 = 0.56). A generalized linear modeling was performed based on this result, considering the response variable (average daily flow) regarding just those significant variables (R2 = 0.49). Subsequently, an inference of models with elimination of variables according to their significance was followed to determine the relative importance of each predictor variable, in order to simplify the model to the minimum number of variables explaining the streamflow. The most significant predictor variable was the mean daily air temperature measured in station 2 ($R^2$ = 0.62), adjusted through a third–order polynomial equation (Eq 11):

MDS T= 4E-5x 3 - 9E-5x 2 - 0x + 0.007 (Eq. 11)

For: MDS T: Tolosa rock glacier conglomerate mean daily streamflow (m3 s-1) x: mean daily air temperature (°C) in station 2

Certain threshold, around 6 °C (Fig. 2b), is needed for a higher flow delivery rate, as

the air temperature increases. The isolation created by the debris layer, which in turn makes a more delayed thermal inertia for the glacier ice thawing (Østrem, 1965; Buk, 2002; Trombotto and Ahumada, 2005), could explain this behavior.

Answer to Referee comment 6.

(1) The discussion of the results can be improved. What has been learned compared to other existing studies (i.e. related to the assessment of drought controls in other regions/climates, or related to normal situation in similar regions?) In its current form it reads more as a summary.

(2) We agree. The discussion was changed to "Results and discussion" as suggested by another referee. It was streamlined and deeply changed. The focus of the discussion is the chemical quantification of each water source and its comparison with the measured streamflow for the glaciated basins, in order to show the importance of groundwater and ice bodies on streamflow. A new hydrological conceptual model was developed; pointing to the importance of doing more studies about the glacier-groundwater-river (stream-aquifer) exchanges (new Fig. 7). Conclusion was changed to "Summary and Conclusions". In that section we recommend a catchment protection were the glacial, periglacial and groundwater inputs are originated, according with the results and the new conceptual model. Also, further work recommendation (concerning residence time analysis) is advised.

(3) New segment of discussion

Based on the results of the PCA, water isotopic composition ($\delta$18O) and electrical conductivity, the variables that mostly explained the two dimensions were used as tracers in an End Member Mixing Analysis. A mixing model with two tracers, allow the identification of three sources. Three water source types contributing to Cuevas River in December were assumed for the EMMA analysis: snow, groundwater and glaciers. The first component to melt will be the snow, which is even observed in Mt. Tolosa rock glacier conglomerate samples (Fig. 5). According to field observations, MODIS

imagery analysis (Fig. 4 and 5) and station 1 data, the snow had already melted for this period before January 2014. Thus, in January and February, it was assumed that the water sources contribution to the Cuevas River changed, being rock glaciers, groundwater and glaciers (Fig. 6). Although glaciers represent 6% of the Cuevas River basin area (IANIGLA–ING, 2018a), during the analyzed melting period, the EMMA analysis shows a contribution between 8 to 9% in the months of January–February, respectively, to the total Cuevas River streamflow, and up to almost 20% in December (Table 5). This high percentage calculated for December may be due to a snowmelt from the upper elevation bands which is not detected with the markers and characterized sources. However, it is more clear that in January and February (when the signal can be purely attributed to ice melt), ice bodies contribute 8.5% (averaging both months). Rock glaciers occupy 17.6 km2 (2.6% of the Cuevas River basin area), but contributes between 47 to 34% of Cuevas River flow in January and February, respectively, according to the EMMA analysis (Table 5). Snowmelt, considered as a contributor only in December under this analysis assumption, contributes 48% to the total flow of the Cuevas River. Groundwater contributes 32, 45 and 56% for the Cuevas River flow rates corresponding to December, January and February, respectively (Table 5). This emphasizes the importance of this source, which may represent either groundwater or the continuous and discontinuous permafrost melting. When analyzing the relative contribution from each glaciated basin to the Cuevas River for each month, the streamflow (Table 4) and the EMMA results (Table 5) both methods yield different information. The streamflow measured at the Tolosa rock glaciers conglomerate represents 0.21 and 0.11% of the Cuevas streamflow for January and February, respectively (Table 4), while the EMMA shows 47 and 34% input from rock glacier source for the same months (Table 5). The small area of this conglomerate may explain the low contributions estimated with direct streamflow measurements. The larger estimates of rock glaciers input estimated with EMMA may be attributed to other rock glaciers draining the Cuevas River. For glacier contributions estimates, considering only the Horcones Inferior Glacier, just one of the 190 crioforms in the basin (IANIGLA-ING, 2018a), the measured streamflow

represents 42.43 and 34.10% (for January and February, respectively) of the Cuevas River streamflow measured in Punta de Vacas (Table 5). For the same months, the EMMA estimates 8 and 9% input from the glacier source (Table 5). In this case, contributions estimated with direct streamflow measurements of one glacier are much larger than the estimated with EMMA, pointing to other processes, not considered with any of the approaches. The water delivered by glaciers might infiltrate through deep fractured aquifers and to the soil, generating a large groundwater matrix draining in the lower basin area, where the Cuevas River flows (Fig. 7). This process may change ion chemistry during water transport to the Cuevas River sampling site, increasing salinity and emerging in EMMA as groundwater source. In addition, between the glacier and the Punta de Vacas Cuevas River sampling site (in Puente del Inca), deep thermal groundwater flow to the river, probably changing its ion concentration. These results imply that the 32, 45 and 56% of groundwater contributions obtained with EMMA for December, January and February, respectively (Table 5), are composed of old glacier water infiltrated to aquifers. In EMMA with two tracers and three components, the three components form the vertices of a triangle and all the river samples must be framed by the triangle. If samples are located outside the triangle, as the Cuevas River in March, it means either that the tracers are not conservative, or there may be contributions from additional sources (Fig. 6). Puente del Inca geothermal waters may represent this additional source (Fig. 1 and 6). The Puente del Inca geothermal waters were confined in a very narrow region of the scatter plot (Fig. 3). This stable isotopes low dispersion may indicate the isolation of water, compared to the surface waters, without being affected by the fluctuations in precipitation or water melting from snow or ice bodies. The deviation to the right of the global meteoric line indicates an enrichment in 18O, probably due to prolonged isotopic exchange with the rocks at temperatures between 25 and 100 °C (Craig, 1963; Aggarwal et al., 2007). Puente del Inca geothermal water presented stable temperatures of 33°C in all samples. According to this hypothesis, the EMMA results for March, where the Cuevas River waters are outside of the triangle, can be explained by the oxygen enrichment caused by the Puente del Inca geothermal

waters input (Fig. 4, 6 and 7). In the Aconcagua River of Chile, situated at the same latitude as the Mendoza River basin, an hydrograph separation, resulting from an EMMA analysis for the 2011-2012 melting period, was done by Rodriguez et al. (2014). They show a December snow contribution of 19-25% to the Juncal River, different from the 48% obtained in this work. In the Chilean work they do not discriminate between the glaciers from the periglacial (rock glaciers) and from the glacial environments. For that large group of glaciarized sources considered in that work, they calculate a contribution of 51-55% in the spring, while for our study it was 19.7% (December month, Table 5). A closer result was observed for subsurface sources, where the Juncal River basin contribution was around 20-30% to the spring flow, and 32% in the Cuevas River. Glacial contributions during the summer increased to 58-66% of the seasonal flow in the Rodriguez et al. study, and in our results were 43-55% (considering glaciers and rock glaciers). The underground sources for the summer were calculated for the Rio Juncal 2011-2012 study at 34-42%, while in our study it was between 45-56%. Although the estimates for both basins differ because of different time period and geographic location of the studies, both point to glaciers as important contributor to river flow.

Summary and conclusions section

Distinct water sources differ in composition along a melting season for both, stable water isotopes and ion chemistry. This reinforces what had been observed at a seasonal scale by Crespo et al. (2016) and allows us to estimate the relative contribution of snow, groundwater, and ice bodies to rivers. The convective summer precipitation events were detected in glaciated basin stream water stable isotope, allowing to pair singular flow increases to particular summer storm events. The snow to ice contribution transition could also be detected in a sub weekly resolution. Air temperature significantly modulates the glaciated basins streamflow along the melting period, presenting a delayed thermal inertia for rock glaciers. Periglacial (rock glaciers) and glacial environments contributions were relevant during this analyzed dry year, with ice bodies and groundwater contributing most to the Cordillera Principal rivers streamflow

in this relatively dry period analyzed. Natural tracers indicated at least, 8-9% of the water was from glacial sources, 34-47% from rock glacier melting and between 32 to 56% from groundwater system. Direct streamflow measurements indicated large discharges for a particular glacier, Horcones Inferior, which was not detected as a proportional glacier contribution downstream at the Cuevas River, with natural tracers. This discrepancy suggests that an important proportion of water derived from glacier melting is infiltrated to groundwater, where it increases ionic composition, and then discharges downstream with the chemical signal of groundwater. Thus, the major contributor to the Cuevas River obtained with natural tracers is groundwater, which may reflect delayed water inputs incoming from glacial and periglacial environments, with some contributions of geothermal groundwater. To estimate groundwater residence times, and validate the conceptual model developed here, other tracers could be used, such as radioactive isotopes (tritium, 14C). This is the first work estimating glacial, periglacial and groundwater contributions to the Mendoza River basin, reinforcing previous assumptions about the importance of ice bodies to maintain river flows. Furthermore, this study points to the importance of glacier-groundwater-river relations, and the need of additional groundwater studies to better map strategic water source areas to be protected, in addition to those included in the National Glacier Inventory.

Please also note the supplement to this comment:
https://www.hydrol-earth-syst-sci-discuss.net/hess-2018-212/hess-2018-212-AC1-supplement.pdf

———————————————————————

**Legend**

| | Horcones Inferior Glacier |
| | Tolosa rock glaciers conglomerate |
| | Vertiente del Inca Spring |
| | Valle Azul Stream |
| | Santa María Stream |
| | Los Puquios Snow Basin |
| | Puente del Inca |
| | Confluencia |
| | Punta de Vacas |

Mt. Aconcagua
6.962 (m asl)

Vacas River

Horcones River

Mendoza River

Cuevas River

Tupungato River

10    0    10    20 km

**Fig. 1.** Map with the digital elevation model, sampling sites and ice bodies. Glacier shapes (marked with red contour) were taken from the glacier official inventory (IANIGLA–ING, 2018a and 2018b).

[Figure]

[Figure]

**Fig. 2.** Scatter plot of streamflow and mean daily air temperature for Horcones Inferior Glacier (A) and for Mt. Tolosa rock glaciers conglomerate (B).

The scatter plot shows δ²H ‰ VSMOW (y-axis, from -120 to -160) versus δ¹⁸O ‰ VSMOW (x-axis, from -22 to -15).

Legend:
- Horcones Inferior Glacier
- Puente del Inca Spring
- Tolosa Rock Glacier
- Vertiente del Inca Spring
- Rivers
- Valle Azul Stream
- Los Puquios Stream
- Santa María Stream
- La Salada Spring
- Confluencia Nueva Spring
- Confluencia Vieja Spring
- Tolosa Snow
- Transition

**Fig. 3.** $\delta$18O and $\delta$2H values scatter plot of the analyzed samples. The adjusted line is the global meteoric water line (Craig, 1961).

[Figure]

**Fig. 4.** Snow covered area (SCA), mean daily streamflow (MDS), mean daily air temperature (MDT) and deuterium composition of the Horcones Inferior Glacier. The black points are the $\delta 2H$ water composition.

[Figure]

**Fig. 5.** Mean daily streamflow (MDS) of Mt. Tolosa rock glaciers conglomerate and snow covered area percentage (SCA). The dots are the deuterium excess values (d‰.

**Emma Cuevas River**

(Dispersion plot showing μS/cm on the y-axis versus δ¹⁸O ‰ on the x-axis, with legend: Glacier, Groundwater, Rock Glacier, Snow Basin, Cuevas Dec., Cuevas Jan., Cuevas Feb., Cuevas Mar.)

**Fig. 6.** Dispersion plot of mean stable water isotope composition and electrical conductivity of different water sources draining to the Cuevas River.

[Figure]

**Fig. 7.** Hydrological conceptual model.

**Table 1:** Mean, standard deviation (SD) and confidence interval (CI) for electrical conductivity of different water sources in comparison with the intercept (Horcones Inferior Glacier). The significant codes are: 0; '***' 0.001; '**' 0.01; '*' 0.05; '.' 0.1. The significant variables are marked in bold.

| Variable | Mean | SD | CI (2.5–97.5%) | p | |
|---|---|---|---|---|---|
| Intercept | 674.1 | 458.9 | -225.41 & 1573.62 | 0.14 | |
| Summer precipitation | -289.1 | 775.1 | -1808.22 & 1230.01 | 0.71 | |
| Rock glacier | 144.3 | 742.2 | -1310.32 & 1598.97 | 0.85 | |
| River | 447.6 | 561.7 | -653.23 & 1548.49 | 0.43 | |
| Valle Azul snow basin | -243.5 | 744.1 | -1701.92 & 1214.96 | 0.74 | |
| Los Puquios snow basin | -527 | 742.4 | -1982.06 & 928.02 | 0.48 | |
| Sta. María basin | -440.3 | 742.6 | -1895.76 & 1015.22 | 0.55 | |
| **Groundwater** | **1495** | **587.6** | **343.29 & 2646.65** | **0.01** | * |

**Fig. 8.** Table 1

**Table 2:** Mean, standard deviation and confidence interval $\delta^{18}O$ composition for water source and the intercept (Horcones Inferior Glacier). The significant codes are: 0; '***' 0.001; '**' 0.01; '*' 0.05; '.' 0.1. The significant variables are marked in bold.

| Variable | Mean | SD | CI (2.5–97.5%) | p | |
|---|---|---|---|---|---|
| Intercept | -20.23 | 0.56 | -21.33-19.13 | $<2e^{-16}$ | |
| **Puente del Inca** | **1.89** | **0.79** | **0.34 & 3.45** | **0.0168** | * |
| **Summer precipitation** | **7.63** | **0.92** | **8.83 & 9.43** | **$<2e^{-16}$** | *** |
| **Rock Glacier** | **3.22** | **0.89** | **1.48 & 4.96** | **0.0003** | *** |
| **Rivers and streams** | **1.43** | **0.68** | **0.09 & 2.76** | **0.0353** | * |
| **Valle Azul snow basin** | **3.07** | **0.91** | **1.28 & 4.86** | **0.0008** | *** |
| **Los Puquios snow basin** | **3.18** | **0.89** | **1.43 & 4.93** | **0.0004** | *** |
| **Santa María basin** | **2.15** | **0.89** | **0.39 & 3.90** | **0.0165** | * |
| **Groundwater** | **1.61** | **0.71** | **0.21 & 3.01** | **0.0238** | * |

**Fig. 9.** Table 2

[Figure]

**Table 3:** Water sources and Cuevas River mean $\delta^{18}O$ ‰ and electrical conductivity.

| Component/Tracer | $\delta^{18}O$ ‰ | CE µS/cm |
|---|---|---|
| Glacier | -20.4 | 762 |
| Groundwater | -18.6 | 2382 |
| Rock glacier | -17.3 | 873 |
| Snow basin | -17.2 | 265 |
| Cuevas River December | -18.3 | 1037 |
| Cuevas River January | -18.1 | 1540 |
| Cuevas River February | -18.3 | 1710 |
| Cuevas River March | -18.02 | 2094 |

**Fig. 10.** Table 3

**Table 4:** Percentage contribution from different kind of ice covered basins to the Cuevas River since December 2013 to March 2014. % of Cuevas River refers to the % of streamflow regarding the Cuevas River. Sources: Cuevas River in Punta de Vacas streamflow: Secretariat of Water Resources, 32.86° S and 69.77°W), Horcones Inferior Glacier (HIG) and Tolosa Rock glacier conglomerate (Tolosa RGC) streamflow were measured in this study.

| | Cuevas River | | HIG | | Tolosa RGC | |
| | $m^3 month^{-1}$ | $Hm^3 month^{-1}$ | $Hm^3 month^{-1}$ | % of Cuevas River | $Hm^3 month^{-1}$ | % of Cuevas River |
|---|---|---|---|---|---|---|
| Dec | 19,316,621 | 19.32 | 8.94 | 46.28 | 0.081 | 0.42 |
| Jan | 15,574,896 | 15.57 | 6.61 | 42.43 | 0.033 | 0.21 |
| Feb | 12,369,370 | 12.37 | 4.22 | 34.10 | 0.014 | 0.11 |
| Mar | 10,984,118 | 10.98 | 2.50 | 22.77 | 0.019 | 0.17 |
| Total | 58,245,005 | 58 | 22 | | 0.15 | |

**Fig. 11.** Table 4

**Table 5:** Percentage contribution from different water sources to the Cuevas River since December 2013 to February 2014, estimated with natural tracers (EMMA).

| Water Source | Dec | Jan | Feb |
|---|---|---|---|
| Glacier | 19.7 | 8 | 9 |
| Groundwater | 32 | 45 | 56 |
| Rock Glacier | 0 | 47 | 34 |
| Snow | 48 | 0 | 0 |

**Fig. 12.** Table 5

**Supplement:**

**Supplementary data**

| Water sources | Places | Sites | Samples |
|---|---|---|---|
| **Rivers** | Vacas, Cuevas, Tupungato and Mendoza rivers in Punta de Vacas. Cuevas River in Puente del Inca. Horcones Superior and Horcones rivers at Mt. Aconcagua Confluencia Camp. | 7 | 42 |
| **Ice bodies** | Horcones Inferior Glacier and Mt. Tolosa rock glaciers conglomerate. | 2 | 34 |
| **Groundwaters** | Vertiente del Inca, La Salada Stream, Confluencia Nueva Spring, Confluencia Vieja Spring and geothermal waters of "Copa de Champagne" and "Viejo Túnel", both in "Puente del Inca". | 6 | 41 |
| **Precipitations** | Collectors at Laguna de Horcones and Confluencia Camp, both in the Mt. Aconcagua Park | 2 | 4 |
| **Snow basins** | Valle Azul, Los Puquios and Santa María | 3 | 33 |

**Table S1** Sampling along the melting period 2013-2014 in Cordillera Principal geological province. Ice body type classification corresponds to the official inventory of glaciers (IANIGLA-ING, 2015a). Sites refers to quantity of sampling sites for each water source

| Station 2 and HI Glacier streamflow | MDS HI m3/s | Soil MDT °C | Air MDT °C | DMaxT °C | DMinT °C | max-min °C |
|---|---|---|---|---|---|---|
| Mean | 2.09 | 7.17 | 4.90 | 3.55 | 0.77 | 2.72 |
| SD | 0.95 | 3.15 | 3.35 | 4.35 | 3.30 | 3.02 |
| VC% | 45.34 | 43.90 | 68.31 | 122.56 | 430.53 | 111.07 |
| Max | 4.88 | 11.60 | 10.68 | 11.27 | 6.41 | 11.27 |
| Min | 0.52 | 1.35 | -3.65 | -5.95 | -7.28 | -5.95 |

| Rock glaciers streamflow | MDS Tolosa m3/s |
|---|---|
| Mean | 0.02 |
| SD | 0.01 |
| VC% | 70.80 |
| Max | 0.05 |
| Min | 0.00 |

| Station 1 | Atm Press hPa | Air MDT °C | DMaxT °C | DMinT °C | RH% |
|---|---|---|---|---|---|
| Mean | 706.44 | 11.12 | 17.82 | 4.74 | 37.27 |
| SD | 1.55 | 2.88 | 3.41 | 2.66 | 17.39 |
| VC% | 0.22 | 25.89 | 19.15 | 56.17 | 46.67 |
| Max | 710.04 | 16.57 | 24.84 | 11.20 | 97.60 |
| Min | 703.11 | 2.89 | 6.25 | -0.53 | 13.60 |

| Station 1 | Soil DMT °C | Wind Dir.° | W mean vel. | W max vel. | SWE (mm) | Incid. Rad. |
|---|---|---|---|---|---|---|
| Mean | 12.58 | 173.07 | 1.87 | 15.83 | 0.00 | 274.66 |
| SD | 2.71 | 55.19 | 1.16 | 3.09 | 0.00 | 54.84 |

| | | | | | | |
|---|---|---|---|---|---|---|
| **VC%** | 21.57 | 31.89 | 62.08 | 19.53 | | 19.97 |
| **Max** | 17.25 | 349.05 | 6.04 | 22.80 | 0.00 | 330.99 |
| **Min** | 6.39 | 16.36 | 0.14 | 7.94 | 0.00 | 64.56 |

**Table S2** Mean values (Mean), standard deviation (SD), % variation coefficient (VC %) and maximum (Max) and minimum values (Min) for stations 2 and 1 weather stations. Mean daily streamflow (MDS) of Horcones Inferior Glacier (HI) and Tolosa rock glaciers conglomerate (Rock glaciers streamflow). Atmospheric pressure in hPa (Atm Press hPa), mean daily air temperature in ℃ (Air MDT), daily maximum and minimum temperature in ℃ (DMaxT and DMinT, respectively), maximum and minimum mean daily temperature (max-min), relative humidity in % (RH%), mean daily soil temperature in ℃ (Soil MDT), wind direction in degrees (Wind Dir º), mean and maximum wind velocity in m/s (W mean/max vel.), snow water equivalent in mm (SWE) and incident solar radiation in W/m$^2$ (Incid. Rad.)

| Site-water source / (mean/range) | $\delta^{18}O$ ‰ | $\delta^2H$ ‰ | d‰ |
|---|---|---|---|
| Cuevas River Pte. del Inca | -17.5/-18.2- to -16.9 | -134.3/-136.5 to -132.2 | 3.9/0.0-8.9 |
| Cuevas River PV | -18.2/-18.8 to -17.9 | -136.8/-141.3 to -134.0 | 6.8/0.0-9.4 |
| Tupungato River PV | -19.3/-19.6 to -18.9 | -144.9/-147.5 to -142.5 | 6.3/0.0-9.9 |
| Vacas River PV | -19.3/-19.5 to -19.1 | -145.5/-147.5 to -143.8 | 7.1/0.0-9.0 |
| Horcones Inferior Glacier | -20.2/-21.0 to -18.8 | -151.1/-157.3 to -139.4 | 10.7/8.2-12.8 |
| Tolosa rock glaciers conglomerate | -17.0/-17.5 to -15.6 | -129.7/-131.6 to -127.6 | 4.2/-3.1 a 8.9 |
| Valle Azul stream | -17.2/-17.6 to -16.1 | -131.8/-133.4 to -129.6 | 2.3/-4.7 a 7.6 |
| Los Puquios snow basin | -17.0/-17.5 to -16.0 | -129.9/-131.6 to -123.4 | 4.9/0.0-8.4 |
| Santa María stream | -18.1/-18.5 to -17.35 | -138.3/-141.3 to -134.8 | 5.12/4.3-8.1 |
| Groundwater | -18.4/-20.2 to -17.4 | -139.8/-154.3 to -132.8 | 6.3/-1.8 a 10.3 |
| Puente del Inca geothermal | -18.3/-18.4 to -18.2 | -141.3/-141.8 to -140.8 | 4.3/4.4-6.26 |
| Summer precipitation | -9.8/-10.9 to -8.7 | -57.3/-66.2 to -48.5 | 20.9/20.7-21.1 |

**Table S3** Mean and range water stable isotopes and deuterium excess (d) composition values for different water sources. PV refers to the samples taken in "Punta de Vacas" site. Pte. del Inca refers to the samples in "Puente del Inca" location

| Site-water source / (mean/range) | (me/L) $Na^+$ | (me/L) $K^+$ | (me/L) $Ca^{+2}$ | (me/L) $Mg^{+2}$ |
|---|---|---|---|---|
| Cuevas River Pte del Inca | 8.67/5.00-11.00 | 0.03/0.00-0.10 | 10.30/8.40-13.80 | 3.67/2.80-5.30 |
| Cuevas River PV | 6.08/2.20-15.00 | 0.05/0.00-0.10 | 9.80/5.70-12.50 | 3.26/1.30-7.40 |
| Tupungato River PV | 3.85/1.30-13.00 | 0.00/0.00-0.00 | 6.33/4.00-15.20 | 2.77/1.90-3.30 |
| Vacas River PV | 0.83/0.70-1.10 | 0.00/0.00-0.00 | 3.64/2.40-4.70 | 1.57/1.20-2.00 |
| Horcones Inferior Glacier | 1.44/0.80-3.00 | 0.05/0.00-1.00 | 4.52/2.30-10.30 | 1.67/0.80-5.30 |
| Tolosa rock glaciers | 0.27/0.10-0.40 | 0.00/0.00-0.00 | 5.86/4.00-8.10 | 2.82/0.60-4.50 |
| Valle Azul stream | 0.36/0.30-0.40 | 0.00/0.00-0.00 | 2.46/1.40-3.60 | 1.35/0.60-2.10 |
| Los Puquios snow basin | 0.80/0.50-1.10 | 0.00/0.00-0.00 | 1.05/0.40-3.00 | 0.26/0.20-1.20 |
| Santa María stream | 0.55/0.20-1.10 | 0.00/0.00-0.00 | 1.41/0.80-2.30 | 0.56/0.30-0.80 |
| Groundwater | 1.29/0.20-10.00 | 0.01/0.00-0.30 | 21.29/3.30-29.60 | 6.29/1.00-10.50 |
| Puente del Inca geothermal | 275.20/110.00-550.00 | 3.50/1.00-5.00 | 47.65/40.50-57.90 | 8.67/2.60-15.60 |

**Table S4** Mean and range cations values for the different water sources

| Site-water source / (mean/range) | (me/L) $HCO_3^-$ | (me/L) $SO_4^{-2}$ | (me/L) $Cl^-$ |
|---|---|---|---|
| Cuevas River Pte del Inca | 2.51/1.97-2.78 | 10.90/9.60-12.10 | 8.20/6.10-11.00 |
| Cuevas River PV | 1.99/1.73-2.44 | 9.51/7.70-14.20 | 5.51/3.10-12.30 |
| Tupungato River PV | 1.87/1.73-2.03 | 6.88/4.40-14.20 | 3.28/1.10-12.10 |
| Vacas River PV | 1.63/1.42-1.84 | 3.63/2.50-5.00 | 0.54/0.30-1.30 |
| Horcones Inferior Glacier | 1.42/0.96-3.55 | 5.33/2.50-11.70 | 0.70/0.10-1.30 |
| Tolosa rock glaciers | 1.74/1.09-2.40 | 6.80/4.40-9.20 | 0.34/0.00-1.70 |
| Valle Azul stream | 1.48/1.13-1.80 | 2.30/1.00-3.30 | 0.27/0.00-1.20 |
| Los Puquios snow basin | 1.42/1.09-2.33 | 0.44/0.10-1.90 | 0.15/0.00-0.70 |
| Santa María stream | 1.47/1.22-1.80 | 0.74/0.10-1.30 | 0.24/0.10-0.70 |
| Groundwater | 3.16/1.84-4.62 | 23.10/1.00-31.30 | 1.13/0.00-7.30 |
| Puente del Inca geothermal | 27.09/23.80-28.11 | 83.70/26.00-343.80 | 213.80/77.20-311.30 |

**Table S5** Mean and range anions values for the different water sources

| Site-water source / (mean/range) | pH | EC (µS/cm) |
|---|---|---|
| Cuevas River Pte del Inca | 7.72/7.68-7.75 | 1839/1473-2205 |
| Cuevas River PV | 8.0/7.80-8.15 | 1461/544-2094 |
| Tupungato River PV | 7.60/6.87-8.03 | 834/626-1031 |
| Vacas River PV | 7.82/7.13-8.10 | 566/391-665 |
| Horcones Inferior Glacier | 7.41/7.41-7.41 | 674/424-1033 |
| Tolosa rock glaciers | 7.74/6.84-8.19 | 809/640-1093 |
| Valle Azul stream | 8.05/7.28-8.90 | 431/286-515 |
| Los Puquios snow basin | 8.12/7.72-8.64 | 144/98-176 |
| Santa María stream | 7.87/6.59-8.41 | 238/121-324 |
| Groundwater | 7.63/6.72-8.60 | 2187/1340-2950 |
| Puente del Inca geothermal | 6.24/6.19-6.29 | 22682/20500-23440 |

**Table S6** pH and electric conductivity (EC) for the different water sources

| Event | Date | MODIS | Rangers | Cache. | Polv. | PV |
|---|---|---|---|---|---|---|
| Light snowfall in Tolosa rock glaciers conglomerate (and something in Horcones Inferior Glacier) areas | November 14$^{th}$ & 19$^{th}$ to 21$^{st}$ 2013 | X | X | | | |
| Light snowfall | November 25$^{th}$ 2013 | X | X | X | | |
| Rain | December 25$^{th}$ 2013 | | X | 10.5 mm | | |
| Light snowfall | January 16$^{th}$ to 18$^{th}$ 2014 | X | X | X | X | X |
| Light snowfall | January 26$^{th}$ to 27$^{th}$ 2014 | X | X | | | |
| Light snowfall | February 14$^{th}$ to 18$^{th}$ 2014 | X | X | X | X | |
| Light snowfall | February 24$^{th}$ & 25$^{th}$ 2014 | X | X | X | X | |
| Light snowfall | March 1$^{st}$ & 2$^{nd}$ 2014 | X | X | X | X | |

**Table S7** Precipitation records from weather stations, MODIS satellite imagery (MODIS) and Mt. Aconcagua Park Rangers (Rangers) daily records. Cache. and Polv. refers to Cacheuta and Polvaredas National Hydric Resources Secretariat weather stations, respectively. PV refers to Punta de Vacas station, dependent on the National Meteorological Service (SMN)

|  | Jul | Aug | Sep | Oct | Nov | Dec | Jan | Feb | Mar | Apr | May | Jun | Mean |
|---|---|---|---|---|---|---|---|---|---|---|---|---|---|
| **MMS 1957-2017** | **21** | **20** | **22** | **28** | **48** | **84** | **101** | **81** | **54** | **34** | **27** | **23** | **45** |
| **1968** | 15 | 15 | 14 | 13 | 19 | 23 | 46 | 56 | 39 | 18 | 15 | 13 | 24 |
| **2004** | 23 | 22 | 24 | 24 | 26 | 43 | 75 | 71 | 46 | 28 | 24 | 23 | 36 |
| **2010** | 20 | 19 | 17 | 19 | 20 | 30 | 44 | 54 | 41 | 27 | 20 | 18 | 27 |
| **2011** | 13 | 13 | 16 | 18 | 29 | 49 | 68 | 58 | 46 | 28 | 20 | 18 | 31 |
| **2013** | **17** | **16** | **16** | **20** | **36** | **73** | **79** | **54** | **36** | **25** | **21** | **18** | **34** |
| **Dry year %** | 85 | 83 | 79 | 66 | 55 | 52 | 62 | 73 | 77 | 74 | 75 | 79 | 72 |
| **2013%*** | **81** | **78** | **72** | **71** | **76** | **87** | **78** | **67** | **67** | **74** | **79** | **79** | **76** |
| **Monthly Hm³** | 56 | 55 | 57 | 76 | 123 | 224 | 270 | 195 | 144 | 88 | 72 | 59 | **1419** |
| **2013 Monthly Hm³** | 45 | 43 | 43 | 54 | 96 | 196 | 212 | 145 | 96 | 67 | 56 | 48 | **1101** |

| **Seasonal contribution** | 1957-2017 Hm³ | Year % | 2013 Hm³ | Year % | 2013** % |
|---|---|---|---|---|---|
| **Jun-Aug** | 169 | 12 | 136 | 12 | 81 |
| **Sep-Nov** | 256 | 18 | 193 | 18 | 75 |
| **Dec-Feb** | **689** | **49** | **552** | **50** | **80** |
| **Mar-May** | 304 | 21 | 220 | 20 | 72 |
| **Total** | **1419** | **100** | **1101** | **100** | **78** |

**Table S8** Mendoza River mean monthly streamflow record (MMS) in $m^3 s^{-1}$ (DGI, 2018) and some extreme dry years (1968, 2004, 2010-11). 2013 %*: refers to 2013 percentage regarding 1957-2017 average. Every year refers to July-June hydrological cycle (p.e.: 2013 year starts in July 2013 and ends in June 2014) obtained from National Secretariat of Hydric Resources (SRH). Long term monthly and seasonal contribution (Hm³ and %) is showed. 2013** refers to the proportion of 2013 seasonal streamflow regarding the 1957-2017 values

| | Mean | Mean | 2004 | 2010 | 2011 | 2013 |
|---|---|---|---|---|---|---|

| | (1987-2015) | (1987-2016) | | | | |
|---|---|---|---|---|---|---|
| **Winter SWE (mm)** | 616 | | 451 | 320 | 432 | **310** |
| **SWE% regarding '87-´15 record** | | | 73 | 52 | 70 | **50** |
| **Mean (Dec- Mar) temperature in ºC** | | 14.5 | 14.7 | 13.7 | 15.1 | **14.6** |
| **Temperature% regarding `'87-'16 record** | | | 101 | 94 | 104 | **101** |

**Table S9** Portillo (3000 m a.s.l.; 32.84ºS – 70.12ºW) winter snow water equivalent (SWE) in mm and summer (December-March) temperature data from El Yeso Embalse Station (2475 m a.s.l.; 33.65ºS – 70.07ºW). The same dry years as Table S8 are showed. Source: Barrios (2018)

| Site | Altitude m asl | Since | Until | $\delta^{18}O$ ‰ | $\delta^{2}H$ ‰ | d‰ |
|---|---|---|---|---|---|---|
| **Confluencia Camp** | 3433 | February 5[th], 2014 | March 15[th], 2014 | -10.9 | -66.2 | 20.7 |
| **Laguna de Horcones** | 3043 | December 19[th], 2013 | April 2[nd], 2014 | -8.7 | -48.5 | 21.1 |
| **Mean** | | | | -9.8 | -57.3 | 20.9 |

**Table S10** Stable water isotopes precipitation (from collectors) composition

[Figure]

**Figure S1:** Historic Mendoza River mean monthly streamflow in $m^3s^{-1}$ (MMS). The mean monthly streamflow from 1957 to 2017 (DGI, 2018) presents the maximum streamflow on January, a 2nd maximum on December and a 3rd one in February, corresponding to a "mitigate glacial regime" according to the Parde genetic classification of solid feeding fluvial regimes (Bruniard, 1994). Many drought periods are plotted below this mean values line, including the 2013 year analyzed in this work. The more extreme dry years in record (1968 and 2010), are also with a maximum discharge month displaced to February, which corresponds to an "ultra-glacial regime", according to the Parde classification

[Figure]

**Figure S2**: Tolosa Rock glaciers conglomerate ice bodies shape, according to the National Glacier Inventory (IANIGLA-ING, 2018a). The purple color indicates active rock glacier facie.The orange, refers to inactive rock glacier facie. Just one of the 3 rock glaciers shows an inactive facie (the left one). The red triangle indicates the streamflow measurement site. Image: Alos

[Figure]

**Figure S3:** Horcones Inferior Glacier shape, according to the National Glacier Inventory (IANIGLA-ING, 2018a). The red triangle indicates the streamflow measurement site. Image: Alos

---

## Author Comment (AC2) · 6 Oct 2018

Responses to Referee 1, identified as follows: (1) comments from Referee, (2) author's response, (3) author's changes in manuscript.

Answer to Referee, comment 1.

(1) An English proofreading must be performed for the manuscript, also including the

figure captures. It is out of scope of this review to address the frequent grammatical deficits or the necessity to rephrase sentences (e.g. "Cuevas, Vacas and Tupungato rivers when join in Punta de Vacas, form the Mendoza River." (P22L7-8) should be "Cuevas, Vacas and Tupungato rivers form the Mendoza River in Punta de Vacas". It is unclear, what "The logo of Copernicus Publications" means in this figure capture.). There are many examples which could be provided here.

(2) We agree. The entire manuscript is being reviewed and corrected by a technical reviewer and native English speaker.

(3) New Figure 1 and caption added. The figure captions were streamlined and the specific indication (P22L7-8) was framed by the legend, becoming no longer necessary in the figure caption. The "The logo of Copernicus Publications" sentence was deleted.

Answer to Referee, comment 2.

(1) The manuscript is generally quite long and it would be helpful to streamline the text and to leave out parts which are not essential.

(2) We agree. The entire manuscript was rewritten. We considered the referee's suggestions, taking out extensive non essential parts of the manuscript.

Answer to Referee, comment 3.

(1) Many measurements were performed and are used in the analysis presented in the manuscript. It would therefore be good to add a section "Study Area and Data Basis" to the manuscript. Here, the authors should add a table summarizing all the measurements performed, which will help the reader to keep an overview. Additionally, the general hydrological characteristics and setting should be described (i.e. long-term mean values of precipitation, discharge, evapotranspiration, temperature etc.). After this overview, it is easier to describe the methods applied, without having to refer to settings of the measurements (as is the case now).

(2) The data description was better explained. Tables with the measured variables were

already included within the Supplementary data (Tables S2 to S6). According to the previous recommendation of "streamline the manuscript", including them in the main text would extend more the paper. We added the long term variables of precipitation, temperature and Cuevas River streamflow (Fig. S1 and Tables S8 and S9) in the Supplementary data.

(3) See Tables S2 to S6, Figure S1 and tables S8 and S9.

Answer to Referee comment 4.

(1) All locations and regions mentioned in the manuscript, figures and figure captions should be consistent (what is currently not the case, e.g. in Fig. 5 and 6; in Fig. 4, for the first time, the Colorado river and Uspallata Stream are mentioned, without further references in the text).

(2) The consistency between text references, figures and figure captions was optimized. In order to give streamline the manuscript, Figures 4, 5 and 6 were not incorporated in the new version.

Answer to Referee comment 5.

(1) In Fig. 1 an overview map is given. Here, substantial improvements are necessary, also in the context of giving the reader a better overview. In the map, measurement locations presented in the table (see (3)) should be displayed. From the map, the topography is not easily understandable – adding a hillshade layer and using different elevation colours would help in this context. The colours and symbology used should be adopted for easier readability (e.g. The boundary colour of the Horocones River Basin is basically the same as the ice bodies). "References" should be "Legend". Other relevant information and locations, which are mentioned in the text, should be included in the map (e.g. Punta de Vacas).

(2) Corrected. The locations and information mentioned in the text were incorporated in Fig. 1. "References" was changed to "Legend".

(3) See new figure 1.

Answer to Referee comment 6.

(1) The quality of the figures needs to be improved. The font sizes are frequently too small and cannot be read easily. The x-axis labels in Fig. 5 & 6 should be consistent. Adding vertical grid lines, e.g. at every month, would help to analyse the temporal dynamics (e.g. begin of snow melt, glacier melt) described in the text. Using colours in Fig. 10, 11 and 12 would be helpful. Also adding a legend in the Fig. 10 and 11 is necessary.

(2) The figures were optimized as recommended. Figures 5 and 6 are not included in the new version.

(3) See new figures 4(ex-10), 5 (ex-11) and 6 (ex-12).

Answer to Referee comment 7.

(1) Figure 3: It is not clear, what is meant with "Ice covered basin efficiency" and "Efficiency related to Cuevas river in %"

(2) Basin efficiency referred to basin yield. "The yield indicates how many times more (or less) water produces a basin per area unit, with respect to the average production of the basin that drains (described before as: "efficiency related to Cuevas River in %"). Figure 3 was deleted. The basin yield regarding the basin area was changed, as suggested in comment 17, to the water contribution, not weighted by area (Table 4).

(3) See Table 4.

Answer to Referee comment 8.

(1) Tab. 1 & 2: No significant variables are marked in bold.

(2) Corrected.

(3) See Tables 1 and 2.

Answer to Referee comment 9.

(1) Tab. 4: Table caption is one of the examples where a rephrasing is needed. The term "rate per month" is not appropriate.

(2) Corrected.

(3) See new Table 5 (ex 4).

Answer to Referee comment 10.

(1) Methodology: Are all sub-sections necessary?

(2) The sub-sections were streamlined.

Answer to Referee comment 11.

(1) The term for "Height-discharge calibration curves" in Hydrology is "rating curves". The authors present on P4L25 and P5L3 equations for Mt. Tolosa rock glacier and the Horcones Inferior Glacier. The equations should be consistently formatted as shown in Eq. 1. In general, I think that these equations do not have an added values for the reader.

(2) The term "Height-discharge calibration curves" was modified to "rating curves". The equations are not longer in the manuscript.

(3) Section 2.1: Streamflow and environmental variables analysis

Two different procedures for each analyzed site were applied for the construction of the rating curves, which links the height measured continuously by the sensor and the volumetric flow. One was by constant saline flow (Gordon et al., 2013) and the other by the velocity–area method (Francou and Poyaud, 2004). Mt. Tolosa rock glaciers conglomerate, composed of four rock glaciers is located between the peaks "Leñas del Tolosa" and the "Morro El Paso", at 32.80°S – 70.01°W (Fig. S2). The rock glaciers range from 3509 to 3749 m a.s.l., with an average altitude of 3614 m a.s.l., a mean

length of 378 m and a total covered area of 0.16 km2. This site was chosen for the simplicity provided by the cryogenic origin of the rock glaciers covering this sub–basin, accessibility throughout most of the year and because it is near ($\sim$ 6 km), and in the same geological province, to the Horcones Inferior Glacier. Furthermore, the similarity in orientation (south) and average altitude of both glaciated basins reduces the environmental noise when comparing water delivery dynamics of these different kinds of ice bodies. Because the Mt. Tolosa rock glaciers conglomerate has a low flow, saline tracer constant flow measurements were performed periodically (Gordon et al., 2013). These dissolution methods involve introducing a tracer substance as a chemical tracer (in this case table salt) in the stream and then monitor concentration changes downstream. This method is especially useful in turbulent mountain streamflow (Moore, 2004; Gordon et al., 2013). The salt solution was injected at a constant flow rate using a Mariotte bottle built for that purpose. Using a digital conductivity meter, calibration curves were constructed with measurements of salinity downstream, after pouring the saline solution. The Horcones Inferior Glacier is located in the Aconcagua Mt. Provincial Park (32.73°S and 69.97°W), drains to the Horcones River, a tributary of the Cuevas River, which drains to the Mendoza River (Fig. 1). The glacier is distributed from 3472 to 5460 m a.s.l., with an average altitude of 4151 m a.s.l. It presents southeast orientation, a total length of 12.7 km and 6.95 km2 area (Fig. S3). To calculate the glacier streamflow, speed–area rating curves were performed (Francoud and Poyaud, 2004).

Answer to Referee comment 12.

(1) P4L26-27: "The determination coefficient between calculated and observed streamflow was 0.95, for y = 0.999x - 2E-05". It is unclear what y = 0.999x - 2E-05 means in this context.

(2) In order to give streamline the manuscript as was recommended, these equations are no longer in the manuscript.

Answer to Referee comment 13.

(1) P5L4: For what is this equation? What is the difference between measured flow (Qm) and calculated flow (Qcalc), since Qm is also calculated as a function of h?

(2) The equation described the fitted values of the simulation with the measured flow, but since the equation is not considered relevant, we eliminated it from the revised draft.

Answer to Referee comment 14.

(1) P6L3-6: Was the data of these totalizer used in the study?

(2) Yes. These data were used to describe the punctual summer storms isotopic enrichment of the Horcones Inferior Glacier streamflow (Fig. 10, now Fig. 4). The isotopic results are shown in Table S3 and the Table S10 was added for more detail. New references were added in the text.

(3) Text references, Figure 4 and Tables S3 and S10.

Text references:

Section 3.2.2 Temporal variability of snow and glacier contributions

The isotopic composition changes observed in the Horcones Inferior Glacier streamflow, from the beginning to the end of the melting season, responds to the contributions of sporadic enriched summer storm events (Fig. 4 and Tables S3 and S10). For example, samples from January 19th and 22nd show enrichments due to summer precipitations on January 16th to18th, which felt as light snow in the Horcones Inferior Glacier area (Fig. 4 and Table S7). The January 29th sample shows the enrichment due to the January 26th and 27th summer storms. Then MODIS imagery indicates that all the snow melted, and glacier contribution yielded streamflow stable isotope impoverishment until February 13rd. Less successive snow melt contributions from January 27th storm were detected until February 19th, due to the February 14th to 18th storm. On February 26th, an enrichment is again registered due to February 24th and 25th storms, with successive impoverishing until the next sampling (3/15/2014), to the mean

values of the proglacial stream. These summer rain or light snow events (Fig. 4 and Tables S3 and S10) increased stable isotope values of streamflow, showing the sensitivity of the measurements.

Answer to Referee comment 15.

(1) P9L6-7: The equations should be formatted as shown in Eq. 1. The naming of the variables should be improved, e.g. TMD air (France) is an odd naming for a variable.

(2) Corrected. All the equations were formatted as recommended and the naming were unified and improved.

(3) New equation:

MDS HI= 0.2221 * MDAT2 + 0.9243 (Eq. 8) MDS HI= 0.2318 * MDAT1 - 0.6489 (Eq. 9)

For:

MDS: Horcones Inferior Glacier mean daily streamflow (m3 s-1)

MDAT2: mean daily air temperature (°C) in station 2

MDAT1: mean daily air temperature (°C) in station 1

Answer to Referee comment 16.

(1) Section 3.1: What other variables were analysed apart from air temperature? I am surprised that the authors do not mention any relationship between glacier melt and solar radiation.

(2) The influence of environmental variables on the streamflow was modeled with the generalized linear effects model, with significant probabilities. The generalized linear effect model was carried out for all the environmental variables measured in stations 1 and 2 (Table S2, already showed in point 3 in C, referee comment 3), expressed in section 2.1. The results for the Horcones Inferior Glacier were expressed in section 3.1.1. The results for the Tolosa rock glaciers conglomerate are expressed in 3.1.2.

(see 3). We also expected solar radiation statistical significance. Solar radiation may influence the ice melting but, during the period studied in the austral summer, the effect was not significant. The temperature (which probably includes the solar radiation effect), was significant.

(3) Text references:

Section 2.1: Streamflow and environmental variables analysis

The Irrigation General Department of Mendoza weather station (labelled as 1 in Fig. 1), located at 3043 m a.s.l. in "Laguna de Horcones" (32.80°S – 69.95°W), measured: air temperature, soil temperature, wind speed and direction, relative humidity, incident radiation and snow water equivalent, hereinafter "station 1". Air and soil temperature HOBO sensors were also installed in the Horcones Inferior Glacier (labelled as 2 in Fig. 1), at 4016 m a.s.l. (32.69°S – 69.97°W), hereinafter "station 2" (Table S2). Both stations covered an altitude gradient of 973 m.

Generalized linear models (nlme package, Pinheiro et al., 2013) in the R program (R Core Team, 2013) were conducted. The streamflow was considered the response variable, as a function of environmental data (predictor variables).

Section 3.1.1: Horcones Inferior Glacier

Streamflow of the Horcones Inferior Glacier showed a similar variability as that of temperatures (Fig. 2a). The best overall linear model obtained for the total measured variables in both, stations 1 and 2 (Table S2), includes as significant variability only for the mean daily air temperature ($p < 0.01$). The Horcones Inferior Glacier average daily streamflow variability, fits linearly with respect to the mean daily air temperature for both stations, following equations 8 ($R2 = 0.7$) and 9 ($R2 = 0.6$), respectively:

MDS HI= 0.2221 * MDAT2 + 0.9243 (Eq. 8)

MDS HI= 0.2318 * MDAT1 - 0.6489 (Eq. 9)

For:

MDS: Horcones Inferior Glacier mean daily streamflow (m3 s-1)

MDAT2: mean daily air temperature (°C) in station 2

MDAT1: mean daily air temperature (°C) in station 1

Section 3.1.2.: Mt. Tolosa rock glaciers conglomerate

From all the environmental variables measured in stations 1 and 2 (Table S2), the most influential variables in the emergent flow of the analyzed rock glacier cluster, were mean daily air temperature (p < 0.01) and mean daily maximum air temperature (p = 0.027), both corresponding to station 2 (R2 = 0.56). A generalized linear modeling was performed based on this result, considering the response variable (average daily flow) regarding just those significant variables (R2 = 0.49). Subsequently, an inference of models with elimination of variables according to their significance was followed to determine the relative importance of each predictor variable, in order to simplify the model to the minimum number of variables explaining the streamflow. The most significant predictor variable was the mean daily air temperature measured in station 2 ($R^2$ = 0.62), adjusted through a third–order polynomial equation (Eq 11):

MDS T= 4E-5x 3 - 9E-5x 2 - 0x + 0.007 (Eq. 11)

For:

MDS T: Tolosa rock glacier conglomerate mean daily streamflow (m3 s-1)

x: mean daily air temperature (°C) in station 2

Certain threshold, around 6 °C (Fig. 2b), is needed for a higher flow delivery rate, as the air temperature increases. The isolation created by the debris layer, which in turn makes a more delayed thermal inertia for the glacier ice thawing (Østrem, 1965; Buk, 2002; Trombotto and Ahumada, 2005), could explain this behavior.

Answer to Referee comment 17.

(1) Section 3.2.1: What does "Glacier covered basins performance" mean? Does the mentioned Cuevas River Basin here refer to Punta de Vacas? Some interesting results, e.g. what is the percentage (not weighted by area) of the contribution of the glacier to the total runoff, is not given here, but can only be found in the discussion. Maybe it would also make sense to merge the results and discussion part.

(2) Glacier covered basins performance indicates how many times more (or less) water produces a basin per area unit regarding the average production of the basin that drains. The basin yield regarding the basin area was changed, as suggested, to the water contribution not weighted by area (Table 4).

The reference to the Cuevas River measured in Punta de Vacas was now incorporated to the manuscript.

The percentages not weighted by area were now included (Table 4) and referred in section 3.1.3 and 3.2. Merging these results with EMMA (Table 5) was very synergetic and the discussion (section 3.2) was enriched (text parts added in point 3).

(3) References in text. See Tables 4 and 5.

Section 3.1.3 Glaciated basins streamflow contributions

The Horcones Inferior Glacier basin streamflow contributes 22 Hm3 along the melting period (Dec-Mar). Tolosa rock glacier conglomerate basin, contributes 0.15 Hm3 for the same period. These basins absolute contributions represent 38 and 0.25% of the Cuevas River streamflow in Punta de Vacas (58 Hm3), respectively (Table 4) for this melting period. The contribution of the Horcones Inferior Glacier is relatively high, considering that it represents only one of the 190 crioforms that discharge to the Cuevas River basin (IANIGLA-ING, 2018a).

Section 3.2
When analyzing the relative contribution from each glaciated basin to the Cuevas River for each month, the streamflow (Table 4) and the EMMA results (Table 5) yield different information. The streamflow measured at the Tolosa rock glaciers conglomerate represents 0.21 and 0.11% of the Cuevas streamflow for January and February, respectively (Table 4), while the EMMA shows 47 and 34% input from rock glacier source for the same months (Table 5). The small area of this conglomerate may explain the low contributions estimated with direct streamflow measurements. The larger estimates of rock glaciers input estimated with EMMA may be attributed to other rock glaciers draining the Cuevas River. For glacier contributions estimates, considering only the Horcones Inferior Glacier, just one of the 190 crioforms in the basin (IANIGLA-ING, 2018a), the measured streamflow represents 42.43 and 34.10% (for January and February, respectively) of the Cuevas River streamflow measured in Punta de Vacas (Table 5). For the same months, the EMMA estimates 8 and 9% input from the glacier source (Table 5). In this case, contributions estimated with direct streamflow measurements of one glacier are much larger than the estimated with EMMA, pointing to other processes, not considered with any of the approaches.

The water delivered by glaciers might infiltrate through deep fractured aquifers and to the soil, generating a large groundwater matrix draining in the lower basin area, where the Cuevas River flows (Fig. 7). This process may change ion chemistry during water transport to the Cuevas River sampling site, increasing salinity and emerging in EMMA as groundwater source. In addition, between the glacier and the Punta de Vacas Cuevas River sampling site (in Puente del Inca), deep thermal groundwater flow to the river, probably changing its ion concentration. These results imply that the 32, 45 and 56% of groundwater contributions obtained with EMMA for December, January and February, respectively (Table 5), are composed of old water infiltrated to aquifers. These old water sources may have originated from snow, glacier or permafrost melting, during unknown periods of time. In EMMA with two tracers and three components, the three components form the vertices of a triangle and all the river samples must be framed by the triangle. If samples are located outside the triangle, as the Cuevas

River in March, it means either that the tracers are not conservative, or there may be contributions from additional sources (Fig. 6). Puente del Inca geothermal waters may represent this additional source (Fig. 1 and 6).

The Puente del Inca geothermal waters were confined in a very narrow region of the scatter plot (Fig. 3). This stable isotopes low dispersion may indicate the isolation of water, compared to the surface waters, without being affected by the fluctuations in precipitation or water melting from snow or ice bodies. The deviation to the right of the global meteoric line indicates an enrichment in 18O, probably due to prolonged isotopic exchange with the rocks at temperatures between 25 and 100 °C (Craig, 1963; Aggarwal et al., 2007). Puente del Inca geothermal water presented stable temperatures of 33°C in all samples. According to this hypothesis, the EMMA results for March, where the Cuevas River waters are outside of the triangle, can be explained by the oxygen enrichment caused by the Puente del Inca geothermal waters input (Fig. 4, 6 and 7).

In the Aconcagua River of Chile, situated at the same latitude as the Mendoza River basin, an hydrograph separation, resulting from an EMMA analysis for the 2011-2012 melting period, was done by Rodriguez et al. (2014). They show a December snow contribution of 19-25% to the Juncal River, different from the 48% obtained in this work. In the Chilean work they do not discriminate between the glaciers from the periglacial (rock glaciers) and from the glacial environments. For that large group of glaciarized sources considered in that work, they calculate a contribution of 51-55% in the spring, while for our study it was 19.7% (December month, Table 5). A closer result was observed for subsurface sources, where the Juncal River basin contribution was around 20-30% to the spring flow, and 32% in the Cuevas River. Glacial contributions during the summer increased to 58-66% of the seasonal flow in the Rodriguez et al. study, and in our results were 43-55% (considering glaciers and rock glaciers). The underground sources for the summer were calculated for the Rio Juncal 2011-2012 study at 34-42%, while in our study it was between 45-56%.

Although the estimates for both basins differ because of a different time period and

geographic location of the studies, both point to glaciers as important contributor to river flow.

Answer to Referee comment 18.

(1) Section: 3.2.2.: The naming of the section needs rephrasing. For what is Fig. 4d, since it is not referred to in the text?

(2) In order to streamline the paper, this old section 3.2.2 was deleted, as Figure 4 and many others.

Answer to Referee comment 19.

(1) Section 3.4: Fig. 13 is not in the manuscript.

(2) Corrected, it referred to figure 11 (now Figure 5).

(3) See Figure 5 (ex-11).

Answer to Referee comment 20.

(1) It would be good to change the "Conclusions" into "Summary and Conclusion" and to add here the main findings and quantitative results from the manuscript, e.g. the contributions of the different water sources to the total flow. These numbers are also missing in the abstract.

(2) Changed "Conclusions" into "Summary and Conclusion" as recommended. Main findings and quantitative results were added in "Summary and Conclusion" and in the abstract.

(3)

Abstract. The Mendoza River flow provides fresh water for more than 1.1 million inhabitants in an agriculture based arid region in the Monte desert. Most of the Mendoza River streamflow derives almost exclusively from winter snow precipitation fell in Cordillera Principal, originated from the Pacific Ocean moisture. In addition to the snow

that precipitates in this area of 3023 km2, there are 951 glaciers, covering an area of 404 km2. Given the high inter–annual variability of snowfall (ranging from 5 to 240% of the long term mean records), strongly affected by ENSO events, and the aridity of the region, it is crucial to quantify the contribution from different water sources to the Mendoza River flow. Glaciers play an important role regulating water availability, with mass accumulation in wet and cold years, and melting in hot, dry years. Combining instrumental records of streamflow from glaciers and rivers, meteorological data, remote sensing of snow covered area and chemical analysis of different water sources, this study attempts to understand the hydrological contribution of different water sources to the Cuevas River, the original tributary of the Mendoza River. Isotopic and ion composition allowed us to differentiate snowmelt from glacier ice melt. In addition, it was possible to detect contributions of summer storms from Atlantic origin that occasionally reach the Cordillera Principal. Finally, with end member mixing analysis, the relative contribution of different water sources was estimated over time, showing the contribution of glacial and periglacial environments as the melting season progressed. Groundwater input to the total flow showed relatively large contributions, 32, 45 and 56% for December, January and February, respectively, pointing to the importance of this water source on maintaining Cuevas River flow.

Section 5: Summary and conclusions

Distinct water sources differ in composition along a melting season for both, stable water isotopes and ion chemistry. These findings reinforce what had been observed at a seasonal scale by Crespo et al. (2016) and allows us to distinguish and estimate the relative contribution of snow, groundwater, and ice bodies to rivers streamflow.

The convective summer precipitation events were detected in glaciated basin streams by a distinct water stable isotope composition, allowing to pair singular flow increases to particular summer storm events. The snow to ice contribution transition could also be detected in a sub weekly resolution. Air temperature significantly modulates the glaciated basins streamflow along the melting period, presenting a delayed thermal

inertia for rock glaciers.

Periglacial (rock glaciers) and glacial environments contributions were relevant during this analyzed dry year, with ice bodies and groundwater contributing most to the Cordillera Principal rivers streamflow in this relatively dry period analyzed. Natural tracers indicated at least, 8-9% of the water was from glacial sources, 34-47% from rock glacier melting and between 32 to 56% from groundwater system. Direct streamflow measurements indicated large discharges for a particular glacier, Horcones Inferior, which was not detected as a proportional glacier contribution downstream at the Cuevas River, with natural tracers. This discrepancy suggests that an important proportion of water derived from glacier melting is infiltrated to groundwater, where it increases ionic composition, and then discharges downstream with the chemical signal of groundwater. Thus, the major contributor to the Cuevas River obtained with natural tracers is groundwater, which may reflect delayed water inputs incoming from glacial and periglacial environments, with some contributions of geothermal groundwater. To estimate groundwater residence times, and validate the conceptual model developed here, other tracers could be used, such as radioactive isotopes (tritium, 14C) and CFCs, SF6, etc. analysis. Those kinds of analysis were out of the scope within this investigation.

The present investigation is the first scientific work estimating glacial, periglacial and groundwater contributions to the Mendoza River basin, reinforcing previous assumptions about the importance of ice bodies to maintain river flows. Furthermore, this study points to the importance of glacier-groundwater-river relations, and the need of additional groundwater studies to better map strategic water source areas to be protected, in addition to those included in the National Glacier Inventory.

Answer to Referee comment 21.

(1) In section 5, the authors write "By deepening our understanding about the delivery and depletion dynamics of different water sources influencing the hydrological

processes of the Mendoza River basin, the vital artery of this arid and fragile territory, these tools are expected to serve to decision makers and to generate the necessary mitigation policies for an improved water resources administration along the actual and future climate change scenarios." (By the way - Word is giving me a warning: "Long sentence (consider revising)"). It is unclear, what is meant with "tools" in this context. What would be an example, how decision makers can generate the necessary mitigation policies for an improved water resources administration along the actual and future climate change scenarios from the results? Are these not empty phrases, which the world anyway has too many?

(2) Modified and simplified as recommended. The whole section was rewritten.

(3) See Section 5 in point 3, Referee comment 20.

Please also note the supplement to this comment:
https://www.hydrol-earth-syst-sci-discuss.net/hess-2018-212/hess-2018-212-AC2-supplement.pdf

**Legend**

| | |
|---|---|
| ▬ | Horcones Inferior Glacier |
| ▬ | Tolosa rock glaciers conglomerate |
| ▬ | Vertiente del Inca Spring |
| ▬ | Valle Azul Stream |
| ▬ | Santa María Stream |
| ▬ | Los Puquios Snow Basin |
| ▮ | Puente del Inca |
| ▲ | Confluencia |
| ○ | Punta de Vacas |

Mt. Aconcagua
6.962 (m asl)

Vacas River

Horcones River

Mendoza River

Cuevas River

Tupungato River

10    0    10    20 km

**Fig. 1.** Map with the digital elevation model, sampling sites and ice bodies. Glacier shapes (marked with red contour) were taken from the glacier official inventory (IANIGLA–ING, 2018a and 2018b).

[Figure]

[Figure]

**Fig. 2.** Scatter plot of streamflow and mean daily air temperature for Horcones Inferior Glacier (A) and for Mt. Tolosa rock glaciers conglomerate (B).

[Figure]

**Fig. 3.** $\delta$18O and $\delta$2H values scatter plot of the analyzed samples. The adjusted line is the global meteoric water line (Craig, 1961).

[Figure]

**Fig. 4.** Time-series showing the snow covered area (SCA), mean daily streamflow (MDS), mean daily air temperature (MDT) and $\delta$2H composition of the Horcones Inferior Glacier (bold black dots).

[Figure]

**Fig. 5.** Mean daily streamflow (MDS) of Mt. Tolosa rock glaciers conglomerate and snow covered area percentage (SCA). The dots are the deuterium excess values (d‰.

[Figure]

[Figure]

**Emma Cuevas River**

Fig. 6. Dispersion plot of mean oxygen stable water isotope composition ($\delta$18O) and electrical conductivity of different water sources draining to the Cuevas River.

[Figure]

**Fig. 7.** Hydrological conceptual model.

**Table 1:** Mean, standard deviation (SD) and confidence interval (CI) for electrical conductivity of different water sources in comparison with the intercept (Horcones Inferior Glacier). The significant codes are: 0; '***' 0.001; '**' 0.01; '*' 0.05; '.' 0.1. The significant variables are marked in bold.

| Variable | Mean | SD | CI (2.5–97.5%) | p | |
|---|---|---|---|---|---|
| Intercept | 674.1 | 458.9 | -225.41 & 1573.62 | 0.14 | |
| Summer precipitation | -289.1 | 775.1 | -1808.22 & 1230.01 | 0.71 | |
| Rock glacier | 144.3 | 742.2 | -1310.32 & 1598.97 | 0.85 | |
| River | 447.6 | 561.7 | -653.23 & 1548.49 | 0.43 | |
| Valle Azul snow basin | -243.5 | 744.1 | -1701.92 & 1214.96 | 0.74 | |
| Los Puquios snow basin | -527 | 742.4 | -1982.06 & 928.02 | 0.48 | |
| Sta. María basin | -440.3 | 742.6 | -1895.76 & 1015.22 | 0.55 | |
| **Groundwater** | **1495** | **587.6** | **343.29 & 2646.65** | **0.01** | * |

**Fig. 8.** Table 1

**Table 2:** Mean, standard deviation and confidence interval $\delta^{18}$O composition for water source and the intercept (Horcones Inferior Glacier). The significant codes are: 0; '***' 0.001; '**' 0.01; '*' 0.05; '.' 0.1. The significant variables are marked in bold.

| Variable | Mean | SD | CI (2.5–97.5%) | p | |
|---|---|---|---|---|---|
| Intercept | -20.23 | 0.56 | -21.33-19.13 | <2e$^{-16}$ | |
| **Puente del Inca** | **1.89** | **0.79** | **0.34 & 3.45** | **0.0168** | * |
| **Summer precipitation** | **7.63** | **0.92** | **8.83 & 9.43** | **<2e$^{-16}$** | *** |
| **Rock Glacier** | **3.22** | **0.89** | **1.48 & 4.96** | **0.0003** | *** |
| **Rivers and streams** | **1.43** | **0.68** | **0.09 & 2.76** | **0.0353** | * |
| **Valle Azul snow basin** | **3.07** | **0.91** | **1.28 & 4.86** | **0.0008** | *** |
| **Los Puquios snow basin** | **3.18** | **0.89** | **1.43 & 4.93** | **0.0004** | *** |
| **Santa María basin** | **2.15** | **0.89** | **0.39 & 3.90** | **0.0165** | * |
| **Groundwater** | **1.61** | **0.71** | **0.21 & 3.01** | **0.0238** | * |

**Fig. 9.** Table 2

**Table 3:** Water sources and Cuevas River mean $\delta^{18}O$ ‰ and electrical conductivity.

| Component/Tracer | $\delta^{18}O$ ‰ | CE µS/cm |
|---|---|---|
| Glacier | -20.4 | 762 |
| Groundwater | -18.6 | 2382 |
| Rock glacier | -17.3 | 873 |
| Snow basin | -17.2 | 265 |
| Cuevas River December | -18.3 | 1037 |
| Cuevas River January | -18.1 | 1540 |
| Cuevas River February | -18.3 | 1710 |
| Cuevas River March | -18.02 | 2094 |

**Fig. 10.** Table 3

**Table 4:** Percentage contribution from different kind of ice covered basins to the Cuevas River since December 2013 to March 2014. % of Cuevas River refers to the % of streamflow regarding the Cuevas River. Sources: Cuevas River in Punta de Vacas streamflow: Secretariat of Water Resources, 32.86° S and 69.77°W), Horcones Inferior Glacier (HIG) and Tolosa Rock glacier conglomerate (Tolosa RGC) streamflow were measured in this study.

| | Cuevas River | | HIG | | Tolosa RGC | |
| | m³month⁻¹ | Hm³month⁻¹ | Hm³month⁻¹ | % of Cuevas River | Hm³month⁻¹ | % of Cuevas River |
|---|---|---|---|---|---|---|
| **Dec** | 19,316,621 | 19.32 | 8.94 | 46.28 | 0.081 | 0.42 |
| **Jan** | 15,574,896 | 15.57 | 6.61 | 42.43 | 0.033 | 0.21 |
| **Feb** | 12,369,370 | 12.37 | 4.22 | 34.10 | 0.014 | 0.11 |
| **Mar** | 10,984,118 | 10.98 | 2.50 | 22.77 | 0.019 | 0.17 |
| **Total** | 58,245,005 | 58 | 22 | | 0.15 | |

**Fig. 11.** Table 4

**Table 5:** Percentage contribution from different water sources to the Cuevas River since December 2013 to February 2014, estimated with natural tracers (EMMA).

| Water Source | Dec | Jan | Feb |
| --- | --- | --- | --- |
| Glacier | 19.7 | 8 | 9 |
| Groundwater | 32 | 45 | 56 |
| Rock Glacier | 0 | 47 | 34 |
| Snow | 48 | 0 | 0 |

**Fig. 12.** Table 5

**Supplement:**

**Supplementary data**

| Water sources | Places | Sites | Samples |
|---|---|---|---|
| **Rivers** | Vacas, Cuevas, Tupungato and Mendoza rivers in Punta de Vacas. Cuevas River in Puente del Inca. Horcones Superior and Horcones rivers at Mt. Aconcagua Confluencia Camp. | 7 | 42 |
| **Ice bodies** | Horcones Inferior Glacier and Mt. Tolosa rock glaciers conglomerate. | 2 | 34 |
| **Groundwaters** | Vertiente del Inca, La Salada Stream, Confluencia Nueva Spring, Confluencia Vieja Spring and geothermal waters of "Copa de Champagne" and "Viejo Túnel", both in "Puente del Inca". | 6 | 41 |
| **Precipitations** | Collectors at Laguna de Horcones and Confluencia Camp, both in the Mt. Aconcagua Park | 2 | 4 |
| **Snow basins** | Valle Azul, Los Puquios and Santa María | 3 | 33 |

**Table S1:** Sampling along the melting period 2013-2014 in Cordillera Principal geological province. Ice body type classification corresponds to the official inventory of glaciers (IANIGLA-ING, 2018a). Sites refers to quantity of sampling sites for each water source.

| Weather Station 2 and HI Glacier streamflow | MDS HI m3/s | Soil MDT °C | Air MDT °C | DMaxT °C | DMinT °C | max-min °C |
|---|---|---|---|---|---|---|
| Mean | 2.09 | 7.17 | 4.90 | 3.55 | 0.77 | 2.72 |
| SD | 0.95 | 3.15 | 3.35 | 4.35 | 3.30 | 3.02 |
| VC% | 45.34 | 43.90 | 68.31 | 122.56 | 430.53 | 111.07 |
| Max | 4.88 | 11.60 | 10.68 | 11.27 | 6.41 | 11.27 |
| Min | 0.52 | 1.35 | -3.65 | -5.95 | -7.28 | -5.95 |

| Rock glaciers streamflow | MDS Tolosa m3/s |
|---|---|
| Mean | 0.02 |
| SD | 0.01 |
| VC% | 70.80 |
| Max | 0.05 |
| Min | 0.00 |

| Weather Station 1 | Atm Press hPa | Air MDT °C | DMaxT °C | DMinT °C | RH% |
|---|---|---|---|---|---|
| Mean | 706.44 | 11.12 | 17.82 | 4.74 | 37.27 |
| SD | 1.55 | 2.88 | 3.41 | 2.66 | 17.39 |
| VC% | 0.22 | 25.89 | 19.15 | 56.17 | 46.67 |
| Max | 710.04 | 16.57 | 24.84 | 11.20 | 97.60 |
| Min | 703.11 | 2.89 | 6.25 | -0.53 | 13.60 |

| Weather Station 1 | Soil DMT °C | Wind Dir.° | W mean vel. | W max vel. | SWE (mm) | Incid. Rad. |
|---|---|---|---|---|---|---|

|        |       |        |       |       |      |        |
|--------|-------|--------|-------|-------|------|--------|
| **Mean** | 12.58 | 173.07 | 1.87 | 15.83 | 0.00 | 274.66 |
| **SD**   | 2.71  | 55.19  | 1.16 | 3.09  | 0.00 | 54.84  |
| **VC%**  | 21.57 | 31.89  | 62.08 | 19.53 |      | 19.97  |
| **Max**  | 17.25 | 349.05 | 6.04 | 22.80 | 0.00 | 330.99 |
| **Min**  | 6.39  | 16.36  | 0.14 | 7.94  | 0.00 | 64.56  |

**Table S2:** Mean values (Mean), standard deviation (SD), % variation coefficient (VC %) and maximum (Max) and minimum values (Min) for weather stations 2 and 1. Mean daily streamflow (MDS) of Horcones Inferior Glacier (HI) and Tolosa rock glaciers conglomerate (Rock glaciers streamflow). Atmospheric pressure in hPa (Atm Press hPa), mean daily air temperature in ºC (Air MDT), daily maximum and minimum temperature in ºC (DMaxT and DMinT, respectively), maximum and minimum mean daily temperature (max-min), relative humidity in % (RH%), mean daily soil temperature in ºC (Soil MDT), wind direction in degrees (Wind Dir º), mean and maximum wind velocity in m/s (W mean/max vel.), snow water equivalent in mm (SWE) and incident solar radiation in $W/m^2$ (Incid. Rad.).

| Site-water source / (mean/range) | δ$^{18}$O ‰ (mean/min-max) | δ$^2$H ‰ | d‰ |
|---|---|---|---|
| Cuevas River Pte. del Inca | -17.5/-18.2- to -16.9 | -134.3/-136.5 to -132.2 | 3.9/0.0-8.9 |
| Cuevas River PV | -18.2/-18.8 to -17.9 | -136.8/-141.3 to -134.0 | 6.8/0.0-9.4 |
| Tupungato River PV | -19.3/-19.6 to -18.9 | -144.9/-147.5 to -142.5 | 6.3/0.0-9.9 |
| Vacas River PV | -19.3/-19.5 to -19.1 | -145.5/-147.5 to -143.8 | 7.1/0.0-9.0 |
| Horcones Inferior Glacier | -20.2/-21.0 to -18.8 | -151.1/-157.3 to -139.4 | 10.7/8.2-12.8 |
| Tolosa rock glaciers conglomerate | -17.0/-17.5 to -15.6 | -129.7/-131.6 to -127.6 | 4.2/-3.1 a 8.9 |
| Valle Azul stream | -17.2/-17.6 to -16.1 | -131.8/-133.4 to -129.6 | 2.3/-4.7 a 7.6 |
| Los Puquios snow basin | -17.0/-17.5 to -16.0 | -129.9/-131.6 to -123.4 | 4.9/0.0-8.4 |
| Santa María stream | -18.1/-18.5 to -17.35 | -138.3/-141.3 to -134.8 | 5.12/4.3-8.1 |
| Groundwater | -18.4/-20.2 to -17.4 | -139.8/-154.3 to -132.8 | 6.3/-1.8 a 10.3 |
| Puente del Inca geothermal | -18.3/-18.4 to -18.2 | -141.3/-141.8 to -140.8 | 4.3/4.4-6.26 |
| Summer precipitation | -9.8/-10.9 to -8.7 | -57.3/-66.2 to -48.5 | 20.9/20.7-21.1 |

**Table S3:** Mean and range of the water stable isotopes and deuterium excess (d) values for different water sources. PV refers to the samples taken in "Punta de Vacas" site. Pte. del Inca refers to the samples in "Puente del Inca" location.

| Site-water source / (mean/range) | (me/L) Na$^+$ | (me/L) K$^+$ | (me/L) Ca$^{+2}$ | (me/L) Mg$^{+2}$ |
|---|---|---|---|---|
| Cuevas River Pte del Inca | 8.67/5.00-11.00 | 0.03/0.00-0.10 | 10.30/8.40-13.80 | 3.67/2.80-5.30 |
| Cuevas River PV | 6.08/2.20-15.00 | 0.05/0.00-0.10 | 9.80/5.70-12.50 | 3.26/1.30-7.40 |
| Tupungato River PV | 3.85/1.30-13.00 | 0.00/0.00-0.00 | 6.33/4.00-15.20 | 2.77/1.90-3.30 |
| Vacas River PV | 0.83/0.70-1.10 | 0.00/0.00-0.00 | 3.64/2.40-4.70 | 1.57/1.20-2.00 |
| Horcones Inferior Glacier | 1.44/0.80-3.00 | 0.05/0.00-1.00 | 4.52/2.30-10.30 | 1.67/0.80-5.30 |
| Tolosa rock glaciers | 0.27/0.10-0.40 | 0.00/0.00-0.00 | 5.86/4.00-8.10 | 2.82/0.60-4.50 |
| Valle Azul stream | 0.36/0.30-0.40 | 0.00/0.00-0.00 | 2.46/1.40-3.60 | 1.35/0.60-2.10 |
| Los Puquios snow basin | 0.80/0.50-1.10 | 0.00/0.00-0.00 | 1.05/0.40-3.00 | 0.26/0.20-1.20 |
| Santa María stream | 0.55/0.20-1.10 | 0.00/0.00-0.00 | 1.41/0.80-2.30 | 0.56/0.30-0.80 |
| Groundwater | 1.29/0.20-10.00 | 0.01/0.00-0.30 | 21.29/3.30-29.60 | 6.29/1.00-10.50 |
| Puente del Inca geothermal | 275.20/110.00-550.00 | 3.50/1.00-5.00 | 47.65/40.50-57.90 | 8.67/2.60-15.60 |

**Table S4:** Mean and range cation concentrations for the different water sources.

| Site-water source / (mean/range) | (me/L) $HCO_3^-$ | (me/L) $SO_4^{-2}$ | (me/L) $Cl^-$ |
|---|---|---|---|
| Cuevas River Pte del Inca | 2.51/1.97-2.78 | 10.90/9.60-12.10 | 8.20/6.10-11.00 |
| Cuevas River PV | 1.99/1.73-2.44 | 9.51/7.70-14.20 | 5.51/3.10-12.30 |
| Tupungato River PV | 1.87/1.73-2.03 | 6.88/4.40-14.20 | 3.28/1.10-12.10 |
| Vacas River PV | 1.63/1.42-1.84 | 3.63/2.50-5.00 | 0.54/0.30-1.30 |
| Horcones Inferior Glacier | 1.42/0.96-3.55 | 5.33/2.50-11.70 | 0.70/0.10-1.30 |
| Tolosa rock glaciers | 1.74/1.09-2.40 | 6.80/4.40-9.20 | 0.34/0.00-1.70 |
| Valle Azul stream | 1.48/1.13-1.80 | 2.30/1.00-3.30 | 0.27/0.00-1.20 |
| Los Puquios snow basin | 1.42/1.09-2.33 | 0.44/0.10-1.90 | 0.15/0.00-0.70 |
| Santa María stream | 1.47/1.22-1.80 | 0.74/0.10-1.30 | 0.24/0.10-0.70 |
| Groundwater | 3.16/1.84-4.62 | 23.10/1.00-31.30 | 1.13/0.00-7.30 |
| Puente del Inca geothermal | 27.09/23.80-28.11 | 83.70/26.00-343.80 | 213.80/77.20-311.30 |

**Table S5:** Mean and range anion concentrations for the different water sources.

| Site-water source / (mean/range) | pH | EC (µS/cm) |
|---|---|---|
| Cuevas River Pte del Inca | 7.72/7.68-7.75 | 1839/1473-2205 |
| Cuevas River PV | 8.0/7.80-8.15 | 1461/544-2094 |
| Tupungato River PV | 7.60/6.87-8.03 | 834/626-1031 |
| Vacas River PV | 7.82/7.13-8.10 | 566/391-665 |
| Horcones Inferior Glacier | 7.41/7.41-7.41 | 674/424-1033 |
| Tolosa rock glaciers | 7.74/6.84-8.19 | 809/640-1093 |
| Valle Azul stream | 8.05/7.28-8.90 | 431/286-515 |
| Los Puquios snow basin | 8.12/7.72-8.64 | 144/98-176 |
| Santa María stream | 7.87/6.59-8.41 | 238/121-324 |
| Groundwater | 7.63/6.72-8.60 | 2187/1340-2950 |
| Puente del Inca geothermal | 6.24/6.19-6.29 | 22682/20500-23440 |

**Table S6:** Mean and range pH and electric conductivity (EC) for the different water sources.

| Event | Date | MODIS | Rangers | Cache. | Polv. | PV |
|---|---|---|---|---|---|---|
| Light snowfall in Tolosa rock glaciers conglomerate (and something in Horcones Inferior Glacier) areas | November 14$^{th}$ & 19$^{th}$ to 21$^{st}$ 2013 | X | X | | | |
| Light snowfall | November 25$^{th}$ 2013 | X | X | X | | |
| Rain | December 25$^{th}$ 2013 | | X | 10.5 mm | | |
| Light snowfall | January 16$^{th}$ to 18$^{th}$ 2014 | X | X | X | X | X |
| Light snowfall | January 26$^{th}$ to 27$^{th}$ 2014 | X | X | | | |
| Light snowfall | February 14$^{th}$ to 18$^{th}$ 2014 | X | X | X | X | |
| Light snowfall | February 24$^{th}$ & 25$^{th}$ 2014 | X | X | X | X | |
| Light snowfall | March 1$^{st}$ & 2$^{nd}$ 2014 | X | X | X | X | |

**Table S7:** Precipitation records from weather stations, MODIS satellite imagery (MODIS) and Mt. Aconcagua Park Rangers (Rangers) daily records. Cache. and Polv. refers to Cacheuta and Polvaredas National Hydric Resources Secretariat weather stations, respectively. PV refers to Punta de Vacas station, dependent on the National Meteorological Service (SMN).

| | Jul | Aug | Sep | Oct | Nov | Dec | Jan | Feb | Mar | Apr | May | Jun | Mean |
|---|---|---|---|---|---|---|---|---|---|---|---|---|---|
| **MMS 1957-2017** | **21** | **20** | **22** | **28** | **48** | **84** | **101** | **81** | **54** | **34** | **27** | **23** | **45** |
| **1968** | 15 | 15 | 14 | 13 | 19 | 23 | 46 | 56 | 39 | 18 | 15 | 13 | 24 |
| **2004** | 23 | 22 | 24 | 24 | 26 | 43 | 75 | 71 | 46 | 28 | 24 | 23 | 36 |
| **2010** | 20 | 19 | 17 | 19 | 20 | 30 | 44 | 54 | 41 | 27 | 20 | 18 | 27 |
| **2011** | 13 | 13 | 16 | 18 | 29 | 49 | 68 | 58 | 46 | 28 | 20 | 18 | 31 |
| **2013** | **17** | **16** | **16** | **20** | **36** | **73** | **79** | **54** | **36** | **25** | **21** | **18** | **34** |
| **2013%*** | **81** | **78** | **72** | **71** | **76** | **87** | **78** | **67** | **67** | **74** | **79** | **79** | **76** |
| **Monthly Hm$^3$** | 56 | 55 | 57 | 76 | 123 | 224 | 270 | 195 | 144 | 88 | 72 | 59 | **1419** |
| **2013 Monthly Hm$^3$** | 45 | 43 | 43 | 54 | 96 | 196 | 212 | 145 | 96 | 67 | 56 | 48 | **1101** |

| Seasonal contribution | 1957-2017 Hm$^3$ | Year % | 2013 Hm$^3$ | Year % | 2013** % |
|---|---|---|---|---|---|
| **Jun-Aug** | 169 | 12 | 136 | 12 | 81 |
| **Sep-Nov** | 256 | 18 | 193 | 18 | 75 |
| **Dec-Feb** | **689** | **49** | **552** | **50** | **80** |
| **Mar-May** | 304 | 21 | 220 | 20 | 72 |
| **Total** | **1419** | **100** | **1101** | **100** | **78** |

**Table S8:** Mendoza River mean monthly streamflow record (MMS) in m$^3$s$^{-1}$ (DGI, 2018). Extreme dry years (1968, 2004, 2010-11) are also shown. 2013 %* refers to the 2013 percentage of the long term mean 1957-2017. Each year refers to July-June hydrological cycle (p.e.: 2013 year starts in July 2013 and ends in June 2014) retrieved from National Secretariat of Hydric Resources (SRH). Long term monthly and seasonal contribution (Hm$^3$ and %) is showed. 2013** refers to the percentage of 2013 seasonal streamflow with respect to the 1957-2017 values.

| | Long term mean (1987-2015) | Long term mean (1987-2016) | 2004 | 2010 | 2011 | 2013 |
|---|---|---|---|---|---|---|
| **Winter SWE (mm)** | 616 | | 451 | 320 | 432 | **310** |
| **SWE% regarding '87-´15 record** | | | 73 | 52 | 70 | **50** |
| **Mean (Dec- Mar) temperature in ºC** | | 14.5 | 14.7 | 13.7 | 15.1 | **14.6** |
| **Temperature% regarding `'87-'16 record** | | | 101 | 94 | 104 | **101** |

**Table S9:** Portillo (3000 m a.s.l.; 32.84ºS – 70.12ºW) winter snow water equivalent (SWE) in mm and summer (December-March) temperature data from El Yeso Embalse Station (2475 m a.s.l.; 33.65ºS – 70.07ºW). The same dry years as Table S8 are showed. Source: Barrios (2018).

| Site | Altitude m asl | Since | Until | δ$^{18}$O ‰ | δ$^{2}$H ‰ | d‰ |
|---|---|---|---|---|---|---|
| **Confluencia Camp** | 3433 | February 5th, 2014 | March 15th, 2014 | -10.9 | -66.2 | 20.7 |
| **Laguna de Horcones** | 3043 | December 19th , 2013 | April 2nd, 2014 | -8.7 | -48.5 | 21.1 |
| **Mean** | | | | -9.8 | -57.3 | 20.9 |

**Table S10:** Stable water isotopes precipitation composition gathered from in-situ collectors.

[Figure]

**Figure S1:** Mendoza River Long term mean monthly streamflow in $m^3 s^{-1}$ (MMS). MMS from 1957 to 2017 (DGI, 2018) presents a maximum in January, a 2[nd] maximum in December and a 3[rd] one in February, corresponding to a "mitigated glacial regime" according to the Parde genetic classification of solid feeding fluvial regimes (Bruniard, 1994). Many drought periods are plotted below this mean values line, including the year 2013 analyzed in this work. The more extreme dry years in the record (1968 and 2010, respectively), show the maximum discharge month displaced to February, which corresponds to an "ultra-glacial regime", according to the Parde classification.

[Figure]

**Figure S2**: Tolosa Rock glaciers conglomerate ice bodies shape, according to the National Glacier Inventory (IANIGLA-ING, 2018a). The purple color indicates active rock glacier facie.The orange, refers to inactive rock glacier facie. Just one of the 3 rock glaciers shows an inactive facie (the left one). The red triangle indicates the streamflow measurement site. Image: Alos.

[Figure]

**Figure S3:** Horcones Inferior Glacier shape, according to the National Glacier Inventory (IANIGLA-ING, 2018a). The red triangle indicates the streamflow measurement site. Image: Alos.